# Transcriptomics Reveals the Putative Mycoparasitic Strategy of the Mushroom *Entoloma abortivum* on Species of the Mushroom Genus *Armillaria*

Rachel A. Koch,[a] Joshua R. Herr[a,b]

aDepartment of Plant Pathology, University of Nebraska, Lincoln, Nebraska, USA
bCenter for Plant Science Innovation, University of Nebraska, Lincoln, Nebraska, USA

**ABSTRACT** During mycoparasitism, a fungus—the host—is parasitized by another fungus—the mycoparasite. The genetic underpinnings of these relationships have been best characterized in ascomycete fungi. However, within basidiomycete fungi, there are rare instances of mushroom-forming species parasitizing the reproductive structures, or sporocarps, of other mushroom-forming species, which have been rarely investigated on a genetic level. One of the most enigmatic of these occurs between *Entoloma abortivum* and species of *Armillaria*, where hyphae of *E. abortivum* are hypothesized to disrupt the development of *Armillaria* sporocarps, resulting in the formation of carpophoroids. However, it remains unknown whether carpophoroids are the direct result of a mycoparasitic relationship. To address the nature of this unique interaction, we analyzed gene expression of field-collected *Armillaria* and *E. abortivum* sporocarps and carpophoroids. Transcripts in the carpophoroids are primarily from *E. abortivum*, supporting the hypothesis that this species is parasitizing *Armillaria*. Most notably, we identified differentially upregulated *E. abortivum* $\beta$-trefoil-type lectins in the carpophoroid, which we hypothesize bind to *Armillaria* cell wall galactomannoproteins, thereby mediating recognition between the mycoparasite and the host. The most differentially upregulated *E. abortivum* transcripts in the carpophoroid code for oxalate decarboxylases—enzymes that degrade oxalic acid. Oxalic acid is a virulence factor in many plant pathogens, including *Armillaria* species; however, *E. abortivum* has evolved a sophisticated strategy to overcome this defense mechanism. The number of gene models and genes that code for carbohydrate-active enzymes in the *E. abortivum* transcriptome was reduced compared to other closely related species, perhaps as a result of the specialized nature of this interaction.

**IMPORTANCE** By studying fungi that parasitize other fungi, we can understand the basic biology of these unique interactions. Studies focused on the genetic mechanisms regulating mycoparasitism between host and parasite have thus far concentrated on a single fungal lineage within the Ascomycota. The work presented here expands our understanding of mycoparasitic relationships to the Basidiomycota and represents the first transcriptomic study to our knowledge that examines fungal-fungal relationships in their natural setting. The results presented here suggest that even distantly related mycoparasites utilize similar mechanisms to parasitize their host. Given that species of the mushroom-forming pathogen *Armillaria* cause plant root-rot diseases in many agroecosystems, an enhanced understanding of this interaction may contribute to better control of these diseases through biocontrol applications.

**KEYWORDS** basidiomycetes, metatranscriptomics, mycoparasitism, plant pathogens, transcriptomics

Address correspondence to Joshua R. Herr, jherr@unl.edu.

**F**ungal mycoparasitism is a nutritional strategy where a living fungus—the host—is parasitized by and acts as a nutrient source for another fungus—the mycoparasite. Certain species of fungi in the Hypocreales (Ascomycota) are among the best-studied mycoparasites. Perhaps the best known of these are species of *Trichoderma* and *Clonostachys rosea*, which have biocontrol activity against plant-pathogenic species of *Botrytis*, *Fusarium*, *Pythium*, and *Rhizoctonia* (1, 2). Other fungal mycoparasites in the Hypocreales include *Tolypocladium* species, many of which are parasites on the reproductive structures, or sporocarps, of species in the genus *Elaphomyces* (Eurotiales, Ascomycota) (3); *Escovopsis weberi*, which is a specialized necrotrophic parasite of fungal gardens of attine ants (4, 5); and *Hypomyces lactifluorum*, which parasitizes the mushrooms of *Russula* species and transforms them into the iconic "lobster mushroom" (6). Within the Basidiomycota, one well-studied genus of mycoparasites is *Tremella*, which contain parasites of corticiaceous basidiomycetes (7) and lichen-forming fungi (8).

Less-studied examples of mycoparasitism involve mushroom-forming fungi that parasitize other mushroom-forming fungi. Fewer than 20 reported mushroom species may engage in this type of interaction, making it an incredibly rare phenomenon given the total number of mushroom-forming fungi (9, 10). Examples of this interaction whereby the parasite does not appear to impact the fitness of the host include *Volvariella surrecta*, which fruits from the pileus of its host, *Clitocybe nebularis* (11), and *Asterophora* species, which colonize their *Russula* or *Lactarius* host after it dies (10). More commonly, though, mushroom mycoparasites deform host sporocarps and likely prevent the dispersal of their spores. *Pseudoboletus parasiticus* fruits from the sporocarps of *Scleroderma* species, which, after infection, are no longer able to mature and disperse their spores (12). *Psathyrella epimyces* causes the deformation of sporocarp tissue of its host, *Coprinus comatus* (13). Additionally, of the 10 mushroom species in the genus *Squamanita*, all are known to be parasites of sporocarps of species in the genera *Cystoderma*, *Galerina*, *Hebeloma*, and *Inocybe* (10).

One of the most frequently encountered putative mycoparasitic interactions between two mushrooms involves species of *Armillaria* and *Entoloma abortivum* (Fig. 1). *Entoloma abortivum* is often encountered fruiting in soil, humus, or decaying logs in deciduous woods (14), while *Armillaria* species are facultative necrotrophs that can cause root rot in forest and agronomic systems worldwide (15, 16). *Entoloma abortivum* was originally described to occur as two morphologies: the mushroom form, which has a pinkish-gray stipe and pileus and pink gills, and then the carpophoroid form, which is generally white and subglobose and does not develop well-formed gills (Fig. 1). Initially, the carpophoroid form was assumed to be *E. abortivum* sporocarps that were malformed due to parasitism by *Armillaria* species (14). However, macro- and microscopic studies of carpophoroid collections determined that they are actually *Armillaria* sporocarps permeated with *E. abortivum* hyphae (17). Laboratory inoculation experiments showed that *E. abortivum* interacts with *Armillaria* sporocarps to disrupt their morphological development (17).

Whether carpophoroids are the result of a mycoparasitic relationship, where *E. abortivum* serves as the mycoparasite and *Armillaria* species serve as the host, remains unknown. To address this, we initiated a transcriptome study after we encountered all three components of this system fruiting in close proximity—carpophoroids and individual sporocarps of *E. abortivum* and *Armillaria*. For a mycoparasite to be successful, there are several crucial steps to the utilization of a fungal host for nutrition. These steps include (i) sensing the host, (ii) recognition and attachment to host hyphae, (iii) initiating defense responses, and (iv) the eventual demise of the host (18). Previous genomic and transcriptomic studies elucidated the genetic machinery that model mycoparasites utilize during each of these steps (18–20). In this work, we used transcriptomic and metatranscriptomic techniques to analyze the genomic toolbox of *E. abortivum* and *Armillaria* during the carpophoroid stage. We show that the gene expression profiles of *E. abortivum* resemble those of known mycoparasitic species, as well as predict certain genes in

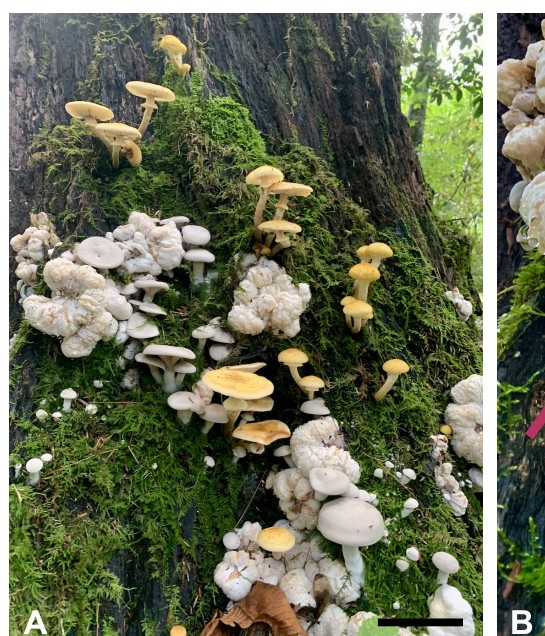
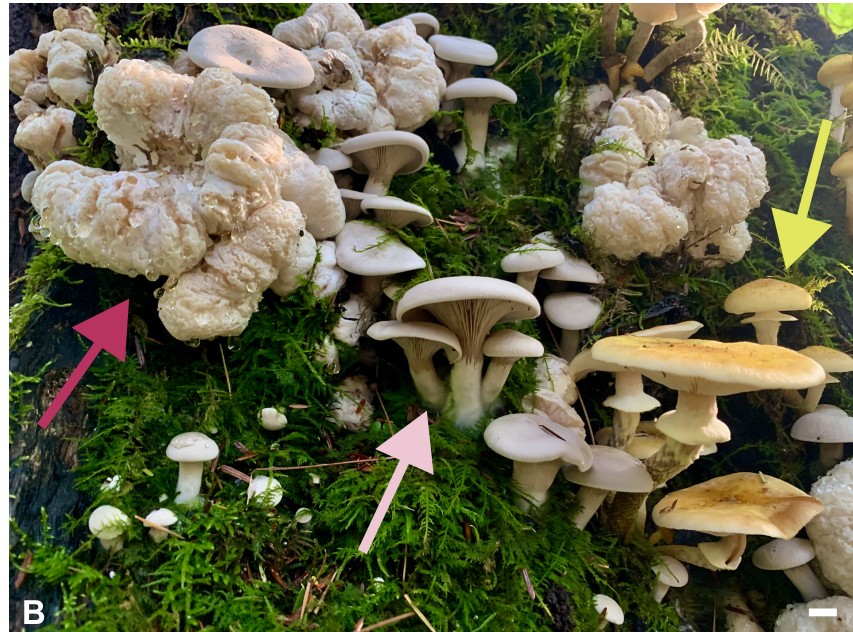

**FIG 1** The components of this fungal interaction in nature. (A) All three components—*Armillaria* and *E. abortivum* sporocarps, along with carpophoroids—fruiting synchronously. (B) A closeup photo of all of the components, with an *Armillaria* sporocarp indicated by the yellow arrow, an *E. abortivum* sporocarp indicated by the light pink arrow, and a carpophoroid indicated by the magenta arrow. Scale bars in panel A are equal to 10 cm, and those in panel B are equal to 1 cm. Photos courtesy of Ben Lemmond, University of Florida (© 2021 Lemmond; reproduced with permission).

both species that facilitate this interaction. Additionally, we used transcriptomic information to determine the species of *Armillaria* involved in this association.

## RESULTS

**Transcriptome assemblies of *Entoloma abortivum* and *Armillaria*.** To benchmark gene diversity and baseline expression levels of the field-collected mushroom species in our study, we sequenced the sporocarp transcriptomes of *E. abortivum* and the *Armillaria* species found in close proximity to the carpophoroids. To date, there are several transcriptomic studies of *Armillaria* species (21, 22), but none for any *Entoloma* species. The assembled transcriptome of *E. abortivum* was just under 120 million bp. There was a total of 43,599 contigs and an $N_{50}$ value of 3,527; 94.5% of benchmark universal single-copy orthologs (BUSCOs) from the Agaricales were present in the *E. abortivum* transcriptome. Nine internal transcribed spacer (ITS) sequences were extracted from the transcriptome, accounting for $5.96 \times 10^{-4}$% to $1.82 \times 10^{-3}$% of the mapped reads; all ITS sequences were from *E. abortivum*. Within the contigs, a total of 9,728 unique gene models were recovered in the transcriptome assembly (Fig. 2) with 603 transcripts differentially upregulated in the carpophoroid tissue compared to the sporocarp tissue and 403 transcripts differentially upregulated in the sporocarp tissue compared to carpophoroid tissue (Fig. 3). The transcriptome contained 195 genes that code for carbohydrate-active enzymes (CAZymes). The transcriptome lacks any genes that code for cellobiohydrolases (glycoside hydrolase [GH] 6 and GH7), xylanases (GH10, GH11, and GH30), and auxiliary proteins like polysaccharide monooxygenases (GH61) but does contain nine chitinases (GH18) (Fig. 2). Transcripts detected in the *E. abortivum* transcriptome that were previously determined to play important roles in mycoparasitic interactions (18–20) include 10 putative secondary metabolite gene clusters, one G-coupled protein receptor (GCPR), 38 ATP-binding cassette (ABC) transporters, and 113 genes from the major facilitator superfamily (MFS) (Fig. 3). The average gene expression (in normalized units of trimmed mean of m-values [TMM]) of the 10 most abundant *E. abortivum* transcripts in the sporocarp ranged from 4,333 to 17,890. Information about the 10 most abundant transcripts in the sporocarps is available in Table 1.

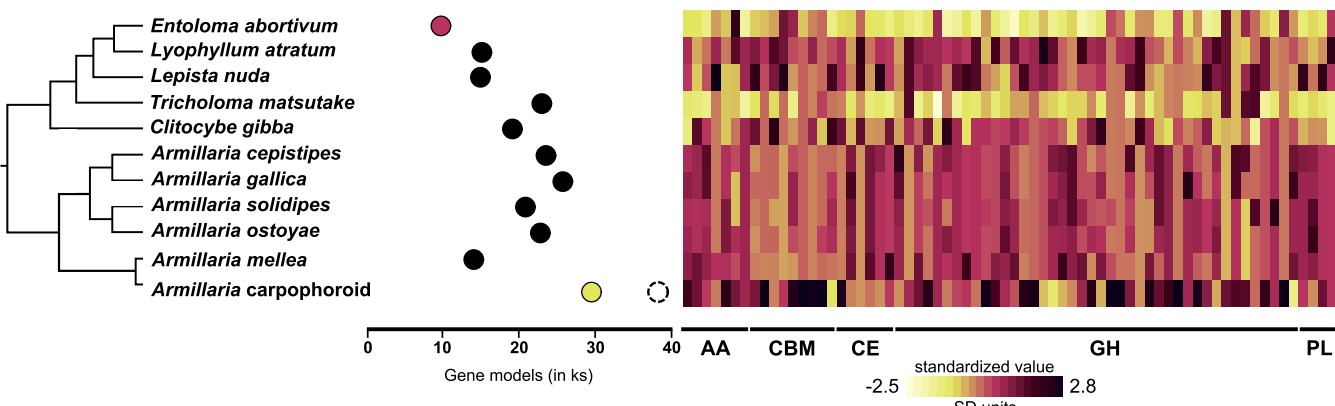

**FIG 2** Number of total gene models and gene copy numbers of CAZymes. (Left) A representative phylogeny based off phylogenetic evidence from references 22 and 111. (Middle) Filled-in, colored dots represent the number of fungal gene models in the transcriptomes of *Armillaria* (yellow) and *E. abortivum* (pink) generated in this study, along with those of closely related species from previously generated genomic data (filled-in black dots). The dashed circle associated with the *Armillaria* transcriptome represents the total number of identified gene models (inclusive of additional non-*Armillaria* genes), showing that nearly one-quarter of the total gene models in the *Armillaria* transcriptome were from organisms other than the target. (Right) Heat map showing gene copy numbers of plant cell wall-degrading enzymes detected in the transcriptomes of *E. abortivum* and *Armillaria*, along with other closely related species. Abbreviations: AA, auxiliary activities; CBM, carbohydrate-binding modules; CE, carbohydrate esterase; GH, glycoside hydrolase; PL, polysaccharide lyase.

The assembled transcriptome of *Armillaria* was just over 138 million bp. A total of 63,905 contigs, an $N_{50}$ value of 2,845, and 97.8% of the BUSCOs representative of the Agaricales were present in the transcriptome. Eleven ITS sequences were extracted from the transcriptome, accounting for $2.77 \times 10^{-5}$% to $6.26 \times 10^{-4}$% of the mapped reads; six ITS sequences were from *Armillaria*, while five were from an uncharacterized species of the yeast *Kodamaea*. A total of 38,215 unique gene models were recovered, and, after bioinformatic filtering against public *Armillaria* genomes, 29,936 of these represented orthologs of *Armillaria* species (Fig. 2). In total, 2,619 transcripts were differentially upregulated in the carpophoroid tissue compared to the sporocarp tissue, whereas 9,820 transcripts were differentially upregulated in the sporocarp tissue compared to the carpophoroid tissue (Fig. 3). The transcriptome contained 580 genes that code for CAZymes, with 34 of those coding for chitinases (Fig. 2). Genes detected in the *Armillaria* transcriptome that might be important in mycoparasitic interactions include 12 putative secondary metabolite gene clusters, five GCPRs, 59 ABC transporters, and 144 MFS transcripts (Fig. 3). The average TMM (normalized units of trimmed mean of m-values) of the 10 most highly upregulated *Armillaria* transcripts in the sporocarp ranged from 8,438 to 36,477. The most differentially upregulated transcript was annotated as a cell wall galactomannoprotein (Fig. 4 and 5; Table 2), while, notably, the 14th most differentially upregulated transcript in the sporocarp coded for isocitrate lyase (Table 2; Fig. 5).

**Metatranscriptomic analysis of combined fungal hyphae in carpophoroid tissue.** We sequenced the metatranscriptome of the mixed tissue in the carpophoroids that are typically found when *E. abortivum* and species of *Armillaria* are found in proximity. In the carpophoroid tissue, significantly more transcriptomic reads from *E. abortivum* were identified than from *Armillaria* sp. ($t_{[8]} = 16.6$, $P = 1.77 \times 10^{-7}$, $n = 9$ replicates per species) (Fig. 6). The average number of *E. abortivum* mapped reads in the carpophoroid tissue was 2,613,988, while the average number of *Armillaria* mapped reads was 74,880 (Fig. 6).

The average TMM of the top 10 most abundant *E. abortivum* transcripts in the carpophoroid ranged from 9,075 to 68,720 (Table 1), while the average TMM of the top 10 most abundant *Armillaria* transcripts in the carpophoroid ranged from 300 to 9,049 (Table 2). The first and third most highly upregulated *E. abortivum* transcripts in the carpophoroid tissue code for two oxalate decarboxylases, both of which were differentially upregulated in the carpophoroid tissue compared to the sporocarp (Fig. 3 to 5 and Table 1). There were other differentially upregulated *E. abortivum* transcripts in the carpophoroid that were not as highly upregulated that may also play a role in

**mSystems**®

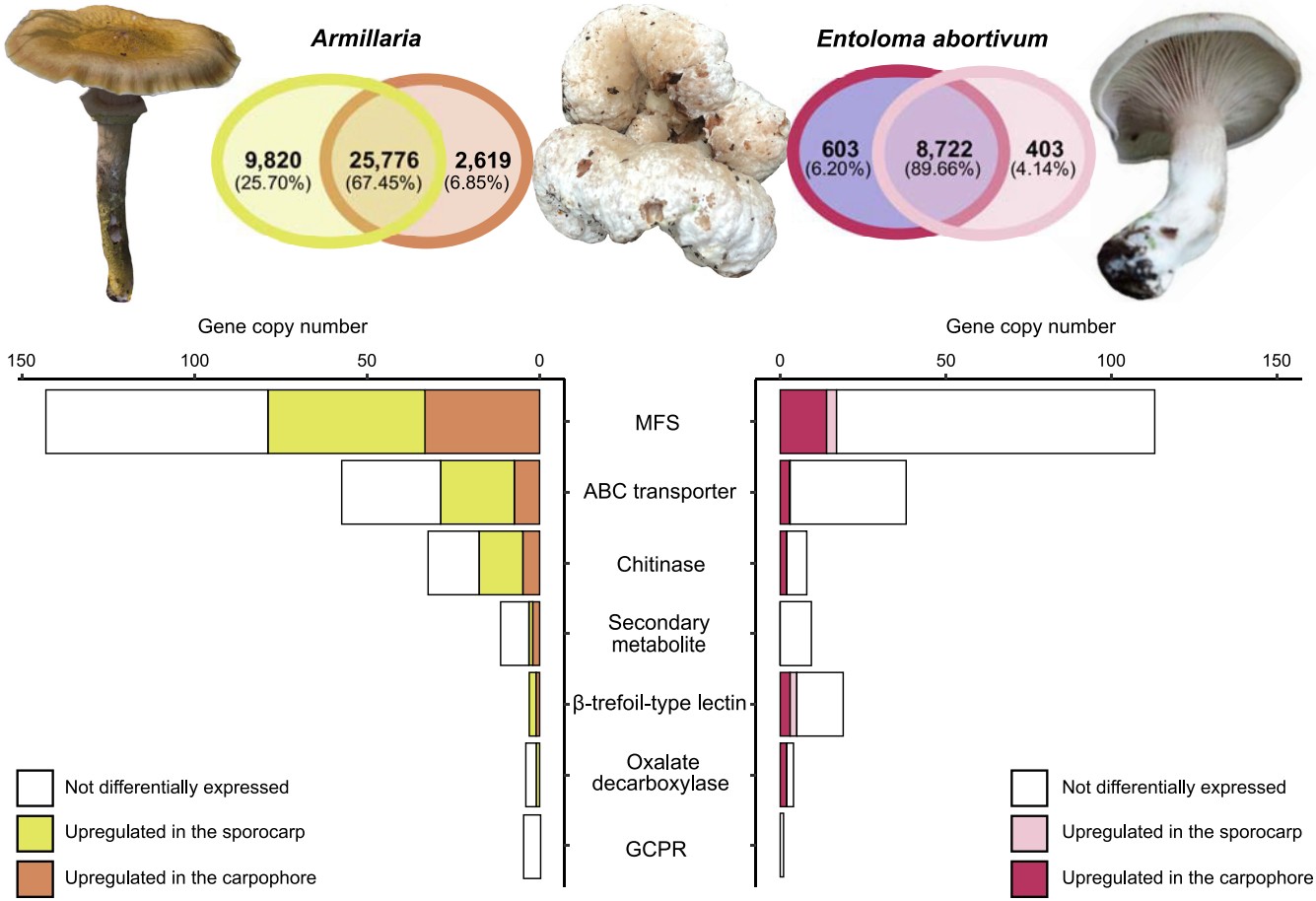

**FIG 3** Differentially upregulated transcripts in the sporocarp and carpophoroid. (Top) Venn diagram showing the number of differentially upregulated transcripts between the sporocarp and carpophoroid in *Armillaria* and *E. abortivum*. Photo of *E. abortivum* courtesy of Eva Skific (© 2021 Skific; reproduced with permission). (Bottom) Bar graph showing the number of copies of genes important in mycoparasitic interactions. The white portion of the bar shows the number of these genes detected in the transcriptome but not differentially upregulated, while the darker colors (brown and purple, respectively) show the number of genes that are differentially upregulated by each species in the carpophoroid and the lighter colors (yellow and coral, respectively) show the number of these genes that are differentially upregulated by each species in their respective sporocarp.

mycoparasitism. These include three *β*-trefoil-type lectins, three ABC transporters, two chitinases, and 14 MFS transcripts (Fig. 3; Table 1). We were unable to detect any transcripts belonging to secondary metabolite gene clusters that were differentially upregulated in the carpophoroids (Fig. 3). Two *Armillaria* transcripts coding for putative senescence-associated proteins, as well as a heat shock protein associated with cell death, were differentially upregulated in the carpophoroid compared to the sporocarp (Table 2).

**Phylogenetic placement of *Armillaria* reads.** Phylogenomic analysis of 100 *Armillaria* BUSCOs generated in this study, in conjunction with previously published *Armillaria* genomes, shows a strongly supported sister relationship with an *Armillaria mellea* specimen from France (100% bootstrap support [BS]) (Fig. 7A). Phylogenetic analysis of all ITS sequences characterized as *A. mellea* from GenBank shows that this specimen is conspecific with specimens from eastern North America (Fig. 7B).

## DISCUSSION

The formation of carpophoroids associated with species of *E. abortivum* has traditionally been thought to be the result of an *Armillaria* species attacking and parasitizing *Entoloma* sporocarps (14), hypothesized on the basis that *Armillaria* species are widespread generalist forest pathogens that have a broad range of host plants (15). However, subsequent studies suggested the opposite: the production of carpophoroids is the result

**TABLE 1** Ten most abundant *E. abortivum* transcripts in sporocarps and carpophoroids[a]

| Gene | Annotation | Cond. | C (TMM) | S (TMM) | logFC | q value |
|---|---|---|---|---|---|---|
| ENT_DN2762_c0_g3 | Oxalate decarboxylase | C | 68,724 | 238 | −8.5 | $4.5 \times 10^{-317}$ |
| ENT_DN1212_c0_g1 | Hypothetical protein | C/S | 55,939 | 12,590 | −2.5 | $4.3 \times 10^{-49}$ |
| ENT_DN3063_c0_g2 | Oxalate decarboxylase | C | 48,157 | 135 | −8.8 | 0 |
| ENT_DN1952_c0_g2 | Acid phosphatase | C | 19,999 | 677 | −5.2 | $1.3 \times 10^{-169}$ |
| ENT_DN4045_c1_g1 | Hypothetical protein | C | 19,965 | 3,374 | −2.9 | $3.4 \times 10^{-63}$ |
| ENT_DN5742_c0_g1 | Hypothetical protein | C | 11,229 | 23 | −9.3 | $6.9 \times 10^{-220}$ |
| ENT_DN4332_c0_g1 | Hypothetical protein | C | 11,038 | 607 | −4.5 | $4.7 \times 10^{-77}$ |
| ENT_DN1952_c0_g1 | Acid phosphatase | C | 10,766 | 673 | −4.4 | $1.9 \times 10^{-122}$ |
| ENT_DN2936_c0_g2 | Hypothetical protein | C | 9,212 | 3,686 | NS | |
| ENT_DN3086_c0_g1 | Hypothetical protein | C | 9,076 | 1,563 | −2.9 | $2.2 \times 10^{-62}$ |
| ENT_DN1375_c0_g1 | Trehalase | S | 1,934 | 17,890 | 2.9 | $6.5 \times 10^{-37}$ |
| ENT_DN5247_c0_g3 | Hypothetical protein | S | 2,718 | 7,204 | NS | |
| ENT_DN761_c0_g2 | Hypothetical protein | S | 7,405 | 6,918 | NS | |
| ENT_DN521_c0_g2 | Hypothetical protein | S | 6,045 | 6,424 | NS | |
| ENT_DN852_c0_g1 | Hypothetical protein | S | 123 | 5,429 | 5.1 | $2.0 \times 10^{-142}$ |
| ENT_DN3621_c0_g2 | RNA polymerase | S | 1,977 | 5,138 | NS | |
| ENT_DN6707_c0_g1 | Hypothetical protein | S | 1,654 | 5,027 | NS | |
| ENT_DN2379_c0_g1 | TPR-like protein | S | 4,862 | 5,017 | NS | |
| ENT_DN466_c0_g1 | Auxin efflux carrier | S | 1,205 | 4,333 | NS | |

[a]Cond., condition under which each gene was most abundant, referring to either the carpophoroids (C) or sporocarps (S). NS, not significantly differentially upregulated. Negative logFC values are significantly differentially upregulated in the carpophoroid, whereas positive values are significantly differentially upregulated in the sporocarp. Transcripts discussed in the text include two oxalate decarboxylases (ENT_DN2762_c0_g3 and ENT_DN3063_c0_g2), three β-trefoil-type lectins (ENT_DN4359_c0_g1, ENT_DN1877_c0_g1, and ENT_DN4255_c0_g1), three ABC transporters (ENT_DN1537_c0_g1, ENT_DN189_c0_g1, and ENT_DN3860_c0_G1), two chitinases (ENT_DN2096_c0_g1 and ENG_DN4507_c0_g3), and 14 MFS transcripts (ENT_DN409_c0_g1, ENT_DN1954_c0_g1, ENT_DN1998_c0_g1, ENT_DN2070_c0_g1, ENT_DN2744_c0_g1, ENT_DN3292_c0_g1, ENT_DN3474_c0_g1, ENT_DN3861_c0_g1, ENT_DN3943_c0_g1, ENT_DN3943_c0_g2, ENT_DN3981_c0_g1, ENT_DN4751_c0_g2, ENT_DN6588_c0_g1, and ENT_DN6695_c0_g1).

of *E. abortivum* disrupting the development of *Armillaria* sporocarps (17). Here, we employed RNA sequencing and differential gene expression analysis on field-collected fungal tissue of each of the three components of this association to better understand the mechanistic basis of this interaction. We determined that *E. abortivum* reads in the metatranscriptome of the carpophoroid tissue—which can be interpreted as a measure of living tissue—are almost 35 times more abundant than *A. mellea* reads (Fig. 6). This finding suggests that carpophoroids are structures that result from *E. abortivum* parasitizing, and eventually killing, its *Armillaria* host under natural conditions.

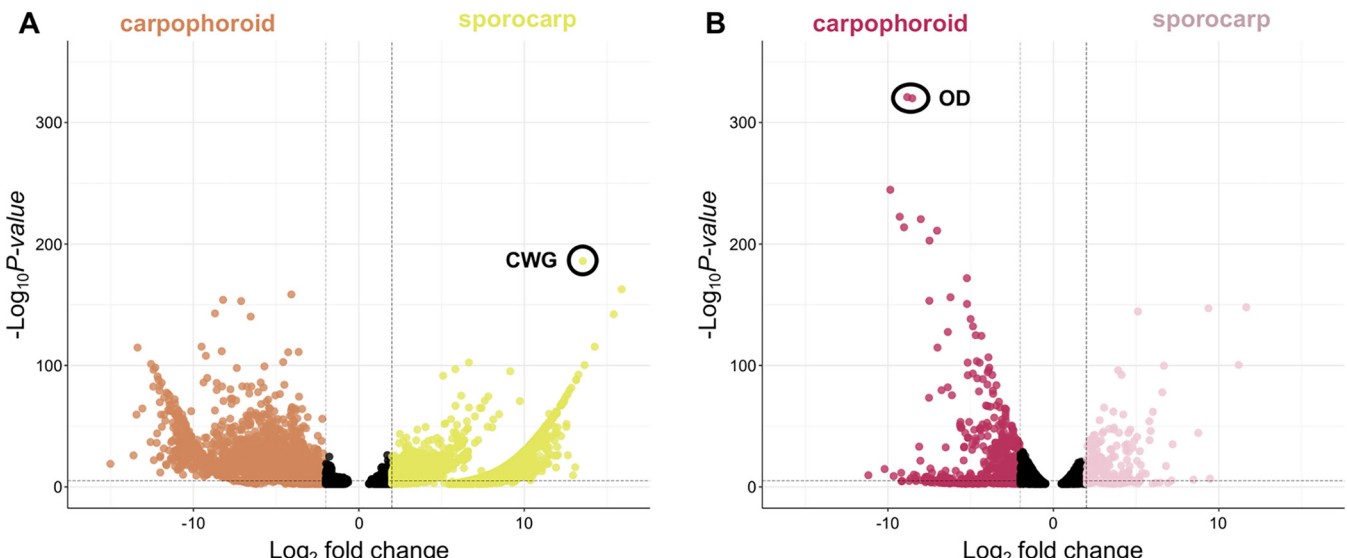

**FIG 4** Volcano plots. Each dot represents a transcript plotted according to its $\log_2$ fold change and the $-\log_{10}$ of its *P* value. Transcripts with the highest statistical significance and the largest fold change are represented by dots toward the top of the plot that are far to either the left (carpophoroid) or right (sporocarp) side. All transcripts with a *q* value of <0.05 are shown. Black dots represent a gene with a nonsignificant logFC ($-2 < \text{logFC} < 2$). (A) *Armillaria*. (B) *E. abortivum*. Dots with a black circle around them are annotated according to the abbreviations: OD, oxalate decarboxylase; CWG, cell wall galactomannoprotein.

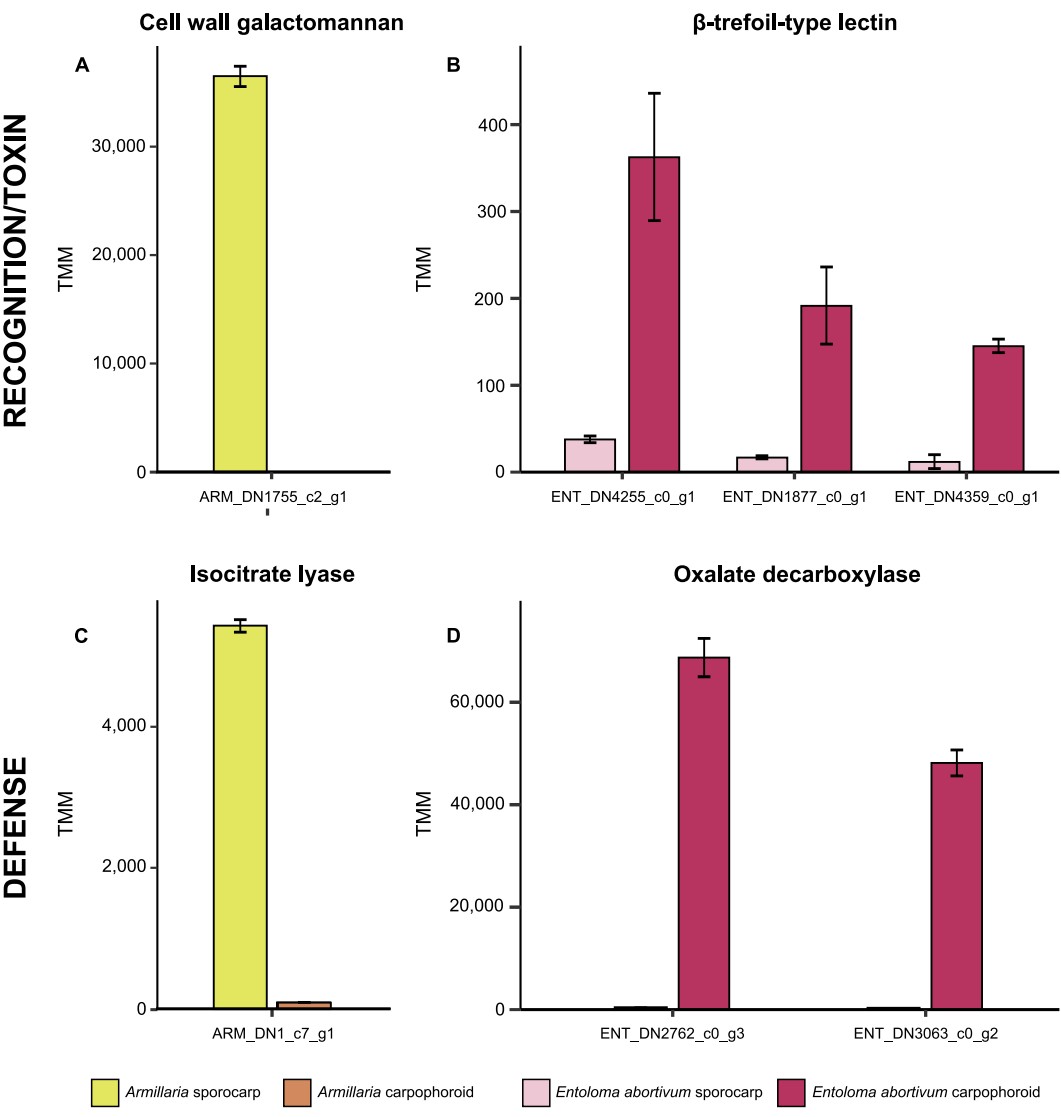

**FIG 5** Genes in both *Armillaria* and *E. abortivum* that are putatively important in the recognition and defense responses during this mycoparasitic interaction. (A) An *Armillaria* cell wall galactomannan that is differentially upregulated in the sporocarp; (B) three *E. abortivum* $\beta$-trefoil-type lectins that are differentially upregulated in the carpophoroid; (C) an *Armillaria* transcript that codes for isocitrate lyase that is differentially upregulated in its sporocarp; (D) two *E. abortivum* transcripts that code for oxalate decarboxylases that are differentially upregulated in the carpophoroids. All results are shown as means $\pm$ standard errors of the means. TMM, normalized units of trimmed mean of m-values.

Fungal-fungal necrotrophic mycoparasitic interactions are multistage processes that are best studied in model species, such as those in the genera *Trichoderma*, *Coniothyrium*, *Clonostachys*, and *Tolypocladium* (20). Genomic and transcriptomic studies of necrotrophic mycoparasites show a convergence of significant genetic mechanisms at each stage (20). The *E. abortivum* genes that are differentially upregulated in the carpophoroid tissue are largely consistent with other examples of necrotrophic mycoparasites in the Ascomycota. In the carpophoroid tissues we analyzed, *E. abortivum* appears to employ much of its energy on recognition and defense responses (Fig. 5). Inversely, the *Armillaria* sporocarps we analyzed illuminate possible mechanisms by which these two species recognize one another and how *Armillaria* responds to parasitism (Fig. 5).

**The genetics of the *Entoloma-Armillaria* mycoparasitic interaction.** A crucial step in a successful mycoparasite's life history is the ability to sense its host. Genes involved in the recognition of the fungal prey include those that code for GPCRs (18, 20).

mSystems®

**TABLE 2** Ten most abundant *Armillaria* transcripts in sporocarps and carpophoroids[a]

| Gene | Annotation | Cond. | C (TMM) | S (TMM) | logFC | q value |
|---|---|---|---|---|---|---|
| ARM_DN1755_c2_g1 | Cell wall galactomannoprotein | S | 0 | 36,479 | 13.5 | $4.0 \times 10^{-182}$ |
| ARM_DN3840_c0_g1 | Serine carboxypeptidase | S | 12 | 29,472 | 6.7 | $7.9 \times 10^{-100}$ |
| ARM_DN22943_c1_g1 | Rab geranylgeranyltransferase | S | 21 | 18,649 | 5.1 | $3.8 \times 10^{-89}$ |
| ARM_DN1737_c0_g1 | LysM-domain-containing protein | S | 0 | 14,557 | 15.9 | $3.0 \times 10^{-159}$ |
| ARM_DN1699_c3_g1 | Hypothetical protein | S | 9 | 12,848 | 5.8 | $1.6 \times 10^{-94}$ |
| ARM_DN1314_c0_g1 | Chondroitin AC/alginate lyase | S | 102 | 11,436 | 2.2 | $3.7 \times 10^{-25}$ |
| ARM_DN20980_c0_g1 | Glycopeptide | S | 0 | 10,449 | 15.4 | $3.4 \times 10^{-139}$ |
| ARM_DN8135_c0_g1 | Rasp f 7 allergen | S | 0 | 9,014 | 11.5 | $1.8 \times 10^{-58}$ |
| ARM_DN4971_c0_g2 | Aldehyde dehydrogenase | S | 103 | 8,688 | NS | |
| ARM_DN1205_c0_g1 | Hypothetical protein | S/C | 9,049 | 8,348 | −4.8 | $2.1 \times 10^{-74}$ |
| ARM_DN5170_c0_g1 | Hypothetical protein | C | 2,086 | 377 | −7.1 | $6.9 \times 10^{-150}$ |
| ARM_DN23207_c0_g2 | Senescence associated | C | 1,711 | 2,482 | −4.1 | $6.6 \times 10^{-61}$ |
| ARM_DN996_c0_g1 | Senescence associated | C | 1,423 | 1,090 | −5.0 | $3.0 \times 10^{-73}$ |
| ARM_DN409_c1_g1 | Elongation factor 1-alpha | C | 769 | 1,154 | −4.1 | $4.0 \times 10^{-155}$ |
| ARM_DN1893_c0_g1 | Hypothetical protein | C | 722 | 58 | −8.3 | $5.3 \times 10^{-109}$ |
| ARM_DN693_c0_g1 | Hypothetical protein | C | 520 | 840 | −3.9 | $1.1 \times 10^{-48}$ |
| ARM_DN1222_c0_g1 | CYS3-cystathionine gamma-lyase | C | 485 | 717 | −4.1 | $5.5 \times 10^{-27}$ |
| ARM_DN1146_c0_g4 | Heat shock protein 70 | C | 431 | 2,361 | −2.2 | $1.7 \times 10^{-54}$ |
| ARM_DN3800_c0_g1 | ATP synthase F1 | C | 300 | 4,686 | NS | |

[a]Cond., condition under which each gene was most abundant, referring to either the carpophoroids (C) or sporocarps (S). NS, not significantly differentially upregulated. Negative logFC values are significantly differentially upregulated in the carpophoroid, whereas positive values are significantly differentially upregulated in the sporocarp. Transcripts discussed in the text were an isocitrate lyase (ARM_DN1_c7_g1), two putative senescence-associated proteins (ARM_DN23207_c0_g2 and ARM_DN996_c0_g1), and a heat shock protein (ARM_DN1146_c0_g4).

However, we did not find any of these genes that were differentially upregulated by *E. abortivum* in the carpophoroid tissue (Fig. 3). Given the significantly lower number of *Armillaria* reads in the carpophoroid tissue compared to *E. abortivum*, we presume that these carpophoroids are relatively advanced in age, and expression of the genes used for sensing the presence of the host is no longer necessary.

We also identified three *E. abortivum* transcripts that code for β-trefoil-type lectins—proteins that bind to galactose units of sugar chains (23)—that were differentially upregulated in the carpophoroid tissue (Fig. 3 and 5). In well-studied mycoparasitic interactions, the recognition, attachment, and coiling around a fungal substrate are mediated by lectins expressed by at least one of the fungal partners (24–26). More specifically, basidiomycete β-trefoil-type lectins have previously been shown to play a role in the recognition of nonself glycans (27). Interestingly, the most abundant and differentially upregulated transcript produced in the *Armillaria* sporocarps, the substrate to which *E. abortivum* hyphae

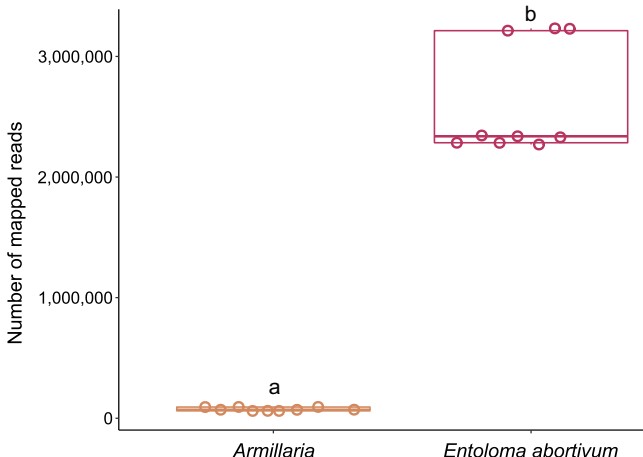

**FIG 6** Boxplots of number of carpophoroid reads that mapped to *Armillaria* and *E. abortivum* when MAPQ (MAPping Quality) was 30. Individual data points are indicated for each species with an open circle. The continuous line within each box represents the mean number of mapped reads. Species labeled with different letters (a to b) have a statistically significant ($P < 0.05$) different number of mapped reads in the carpophoroid.

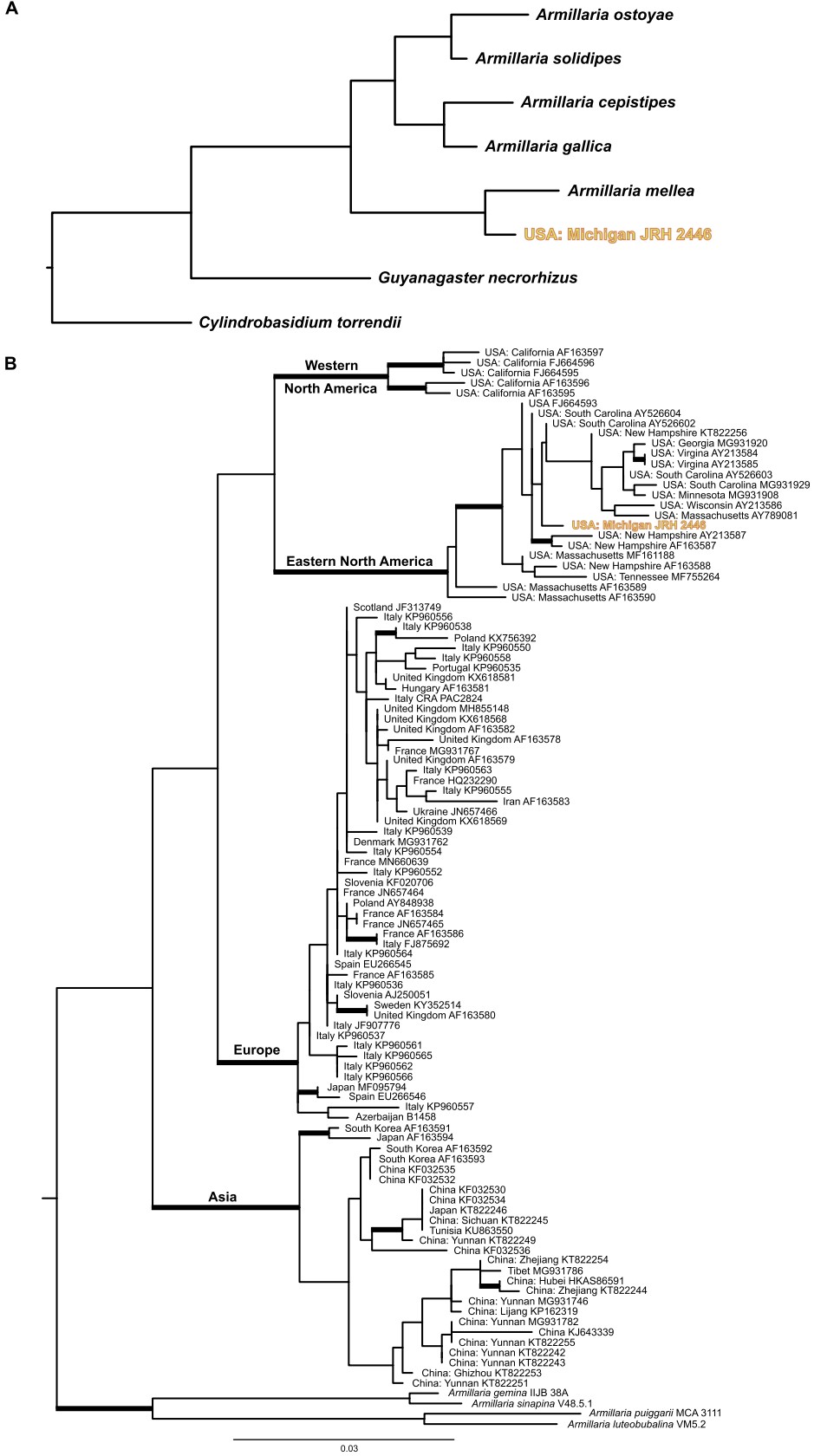

**FIG 7** Phylogenetic placement of the *Armillaria* species used in this analysis. (A) Maximum likelihood phylogeny of *Armillaria* species and *Guyanagaster necrorhizus* generated from the analysis of 100 random

attach, codes for a cell wall galactomannoprotein (Fig. 4 and 5). These proteins belong to a group of glycans which consist of a mannose backbone with galactose side chains and are known to make up a major part of the cell wall of some fungal species (28). This particular galactomannoprotein appears to be specific to *Armillaria* species. Watling (14) commented on the highly specific nature of this interaction and that it has been documented only occurring between *E. abortivum* and *Armillaria* species. One possible mediator of the specificity of this interaction could be the galactose sugars on the mannose protein (known thus far only from *Armillaria* species) that are the means by which *E. abortivum* β-trefoil-type lectins recognize and attach to the *Armillaria* host. However, more genome sequencing of other Agaricales species is needed to determine whether this protein is truly specific to species in the genus *Armillaria*.

During mycoparasitic interactions, the fungal host responds by mounting its own defense, and a successful mycoparasite must be able to cope with this counterattack (18, 20). Oxalic acid (OA) is a virulence factor employed by some plant pathogens, including species of *Armillaria*, to compromise the defense responses of the host plant by creating an acidic environment (29, 30). One differentially upregulated *Armillaria* transcript in the sporocarps codes for isocitrate lyase (Fig. 5), which is involved in OA biosynthesis in other fungal pathogens (31, 32), and we suspect that *Armillaria* sporocarps may utilize OA to defend against the initial infection by *E. abortivum*. However, two of the most abundant and differentially upregulated *E. abortivum* transcripts in the carpophoroid tissue code for oxalate decarboxylases—enzymes responsible for the degradation of OA (Fig. 3, 4, and 5D; Table 1). In at least one known well-studied mycoparasitic interaction, OA is secreted by the fungal host, *Sclerotinia sclerotiorum*, in reaction to penetration by its mycoparasite, *Coniothyrium minitans*. The acidic environment created by the secreted OA inhibits conidial germination and suppresses mycelial growth of *C. minitans* (33). However, *C. minitans* nullifies the growth-suppressing effects of OA or OA-mediated low pH by degrading the OA (34, 35), an enzymatic process largely mediated by oxalate decarboxylase. Because of this, oxalate decarboxylase plays an imperative role in mycoparasitism as OA degradation is vital for infection of the fungal host (36). Given the abundance of these genes produced by *E. abortivum* in the carpophoroid, we suspect a similar scenario in this mycoparasitic interaction. Subsequent downregulation of isocitrate lyase by *Armillaria* in the carpophoroids suggests that its genetic defense responses to infection were likely over at the time of our sampling.

Other ways that mycoparasites cope with the counterattack launched by their host include actively excreting host-secreted toxins. Here, we hypothesize that active extrusion of toxins secreted by the host occurs in the *E. abortivum* carpophoroid tissue via membrane transporters in the ABC superfamily (37–39). Three ABC transporters were differentially upregulated in the carpophoroid tissue (Fig. 3). Another group of genes that were differentially upregulated by *E. abortivum* in the carpophoroid belong to the major facilitator superfamily (MFS) transporters (Fig. 3). In *C. rosea*, there was selection for genes in this family that were related to drug resistance and the transport of secondary metabolites, small organic compounds, and carbohydrates (40). Their importance to mycoparasitism in *C. rosea* is predicted to invoke efflux-mediated protection against exogenous or endogenous secondary metabolites and nutrient uptake (40). MFS transporters have also been shown to be induced in other mycoparasitic species (19, 41), but their exact biological roles have not been investigated.

In mycoparasitism, the final death of the host often results from the synergistic actions of cell wall-hydrolytic enzymes and antifungal secondary metabolites (18, 20).

**FIG 7** Legend (Continued)
BUSCOs. The outgroup taxon is *Cylindrobasidium torrendii*. Each node is fully supported with 100% bootstrap support. (B) Maximum likelihood phylogeny of *Armillaria mellea* phylogenetic tree generated from the analysis of the ITS region. Branches with 70% or more bootstrap support are thickened. Outgroup taxa include *A. gemina*, *A. luteobubalina*, *A. puiggarii*, and *A. sinapina*. The *Armillaria* specimen analyzed during this study is in bold and yellow in both phylogenies.

No secondary metabolite gene clusters identified in the *E. abortivum* transcriptome were differentially upregulated in the carpophoroid tissue (Fig. 3). In some mycoparasitic relationships, the secretion of secondary metabolites occurs early in the interaction, including in *Escovopsis weberi*, which secretes toxic compounds that kill the leaf-cutter ant garden before contact (5). In culture experiments between *Armillaria* isolates and *E. abortivum*, the growth of *Armillaria* was severely inhibited by the presence of *E. abortivum* (42). This suggests that *E. abortivum* may potentially secrete a toxic compound early in the interaction that inhibits the growth of *Armillaria*. Given that significantly more of the living tissue in the carpophoroids belonged to *E. abortivum* (Fig. 6), it is possible that much of the *Armillaria* tissue was killed preceding the full development of the carpophoroid.

Additionally, we hypothesize that the upregulated $\beta$-trefoil-type lectin in *E. abortivum* that may be important in hyphal recognition may also be cytotoxic toward *Armillaria*. This type of lectin has sequence homology, as well as putative structural similarity, to the B-subunit of ricin, a toxic protein from the castor bean *Ricinus communis* (43). An array of $\beta$-trefoil-type lectins have been characterized from the sporocarps of the mushroom-forming species *Clitocybe nebularis* (44), *Coprinus cinerea* (27), *Macrolepiota procera* (45), and *Boletus edulis* (46). Besides being important for nonself recognition, these same mushroom lectins also exhibit entomotoxic activity (47) as well as nematoxic activity (27, 45, 48). Taken together, it is possible that the *E. abortivum* $\beta$-trefoil-type lectins may also function as toxins toward *Armillaria*. While the *E. abortivum* transcripts coding for these lectins are not in the highest abundance in the carpophoroid tissue (Fig. 5), this could be because most of the *Armillaria* sporocarp tissue is already dead and the potential lethal effects produced by them are no longer necessary.

Chitin is an essential polymer in fungal cell walls (49) and is an important target during mycoparasitic attack (20). Indicative of the importance of chitinases in mycoparasitic interactions, members of the genus *Trichoderma*, as well as *Tolypocladium ophioglossoides* and *Escovopsis weberi*, have an increased number of genes coding for them (19, 50–53). Nine fungal chitinases were detected in the transcriptome of *E. abortivum*, which is fewer than the 13, 19, 20, and 29 detected in the closely related species *Tricholoma matsutake* (54), *Clitocybe gibba* (55), *Lyophyllum atratum* (56), and *Lepista nuda* (55), respectively (Fig. 2). Only two of those *E. abortivum* chitinases were differentially upregulated in the carpophoroid tissue (Fig. 3) and were not abundant in comparison to other genes, suggesting minimal significance at this stage in carpophoroid development. One possibility for this difference in abundance could be the result of the putatively acidic pH in the carpophoroid that we infer based on the high gene expression of oxalate decarboxylases. In *C. minitans*, chitinase activity is positively correlated with ambient pH ranging from 3 to 8 (57), so it is possible that chitinase activity in *E. abortivum* will increase after a neutral pH is restored. Another possibility for the low abundance is that most of the *Armillaria* host tissue was already broken down, reducing the need for chitinase activity.

Some of the putatively mycoparasitism-related genes outlined above were also differentially upregulated by *Armillaria* in the carpophoroid tissue. These include genes that code for MFS, ABC transporters, chitinases, and secondary metabolite gene clusters (Fig. 3). This suggests that *Armillaria* may be using many of the same genetic mechanisms to defend itself against parasitism by *E. abortivum*. Additionally, the degree of expression changes—in both the number of differentially upregulated transcripts and the log fold change (logFC)—between the sporocarp and carpophoroid is much greater in *Armillaria* than in *E. abortivum* (Fig. 3 and 4), which could reflect an increase in the level of defense from *Armillaria*. However, this defense is apparently not enough to overcome the parasitic adaptations of *E. abortivum*.

**Gene and CAZyme content of *E. abortivum*.** The number of predicted gene models in the transcriptome of *E. abortivum* was 9,728, which is markedly fewer than the number of gene models in the genomes of its closest sequenced relatives (Fig. 2,

middle panel). Additionally, relative to other closely related mushroom species, *E. abortivum* also exhibits a strong reduction in several gene families encoding CAZymes (Fig. 2, right panel) and contains no cellobiohydrolases, xylanases, or polysaccharide monooxygenases. This finding is consistent with what has been observed in obligate mycoparasites and animal pathogens, which also have reduced CAZyme repertoires compared to fungi that utilize other nutritional strategies, hypothesized to be the result of their highly specialized interactions with their hosts (50, 58, 59). Therefore, it is possible that *E. abortivum* retained only the CAZymes and accessory genes necessary to interact with *Armillaria* species. While a genome sequence of *E. abortivum* will be necessary to confirm this reduction, the BUSCO analysis verified that the *E. abortivum* transcriptome contains nearly 95% of the core set of eukaryotic genes, which suggests that our transcriptome is relatively complete, and it is unlikely that entire gene families are underrepresented.

Another possibility for the reduction in CAZymes could be explained by a broader nutritional strategy employed by *Entoloma* species, some of which form ectomycorrhiza-like structures on host plant species (60–62). Ectomycorrhizal species have a marked reduction in CAZymes in comparison to their saprotrophic ancestors (63), which we also observe with *Tricholoma matsutake* (Fig. 2, right panel). However, microscopic analyses of *Entoloma* ectomycorrhiza-like structures suggest that some species destroy root meristems and young root cells, suggestive of a more parasitic relationship (60, 61). One explanation is that *Entoloma* species, in general, are parasites of true ectomycorrhizae. More broadly, other species within the Entolomataceae are suspected mycoparasites, as they have been reported growing in close association with the sporocarps of other species (reviewed in reference 17), including *Entoloma parasiticum* (=*Claudopus parasiticus*) and *Clitopilus fasciculatus* (64, 65). This explanation would also add credence to the evidence that Entolomataceae species are difficult to culture and are slow growing (62). Additional research utilizing genomes and laboratory studies to understand the nutritional strategy employed by this lineage will inform us as to whether fungal parasitism in this group is more common than it is currently understood to be. Lastly, it should be noted that *Entoloma* species, and *E. abortivum* in particular, form sclerotia in culture and presumably in soil (14, 17). These resting structures are perhaps a dormancy mechanism in the soil to survive seasonality when host tissue is unavailable. Because the timing and presence of *E. abortivum* fruitings have been hard to predict, we were not able to observe or acquire a transcriptome for the sclerotia in the field.

**Gene content and identity of *Armillaria* species in this interaction.** The number of predicted gene models in the transcriptome of this *Armillaria* species was 38,215 (Fig. 2, middle panel). This is substantially greater than all other sequenced *Armillaria* species, which range from 14,473 to 25,704 gene models (22, 66). However, when we looked for gene models in our transcriptome that belong to known *Armillaria* species, this reduced the total number of gene models to 29,936 (Fig. 2, middle panel). The excess gene models in the *Armillaria* transcriptome, compared to reference genomes, likely represent duplicated gene models with splice variants, a common artifact of transcriptome sequencing (67). Additionally, nearly one-quarter of the gene models in the *Armillaria* sporocarps were from organisms other than *Armillaria*, including the yeast *Kodamaea*, highlighting the fact that field-collected sporocarps are not composed of tissue from a single organism. However, none of these contaminating organisms had genes that were both abundant and differentially upregulated that we predicted to play a role in this interaction.

Phylogenomic analysis of the *Armillaria* transcripts generated in this study suggests that the specific *Armillaria* species parasitized in this relationship is sister to an *A. mellea* specimen collected from western Europe (Fig. 7A). An ITS-based phylogenetic analysis shows the *Armillaria* specimen collected in this study is conspecific with other *A. mellea* collections from eastern North America (Fig. 7B). Before now, observations of *A. mellea sensu stricto* fruiting in proximity to carpophoroids (17) hinted that it may be a

host to *E. abortivum*. Here, though, we show for the first time using genomic data that *A. mellea sensu stricto* can definitively serve as a host for *E. abortivum*. However, our hypothesis is that this interaction does not appear specific to just *A. mellea sensu stricto* as *Armillaria gallica*, *Armillaria ostoyae*, *Armillaria jezoensis*, *Armillaria* sp. Nag. E, and *Desarmillaria tabescens* have been previously confirmed as hosts in this interaction as well (17, 42, 68). However, a broad geographic sampling of carpophoroids using molecular markers or genomic information could address this question more thoroughly. Interestingly, *Armillaria* species parasitized by *E. abortivum* appear to be only those present in eastern North America and eastern Asia (17, 42, 68).

**Conclusions.** Data from this study support the hypothesis that *E. abortivum* is a mycoparasite of *Armillaria* sporocarps. Three $\beta$-trefoil-type lectins are differentially upregulated by *E. abortivum* in the carpophoroid tissue, and we propose that these lectins mediate recognition with *Armillaria* sporocarps through binding to an *Armillaria*-specific galactomannoprotein. We hypothesize that by using oxalate decarboxylase, *E. abortivum* is likely defending against the secretion of OA by *Armillaria*. These strategies employed by *E. abortivum* for recognition and defense are similar to mechanisms utilized by other mycoparasites, suggesting that even distantly related mycoparasites utilize similar genetic mechanisms to mediate mycoparasitic interactions. One weakness of this study is that we were limited to the carpophoroid life stage that was available at the point of collection, which led us to speculate about what is occurring during other stages of mycoparasitism (i.e., sensing the host, initiating the interaction, and killing and consuming the host). Therefore, future studies using culture methods, isotopic analysis, and metatranscriptomics of naturally collected carpophoroids at different life stages (i.e., younger and older carpophoroid specimens) will be necessary to completely tease apart the putative mycoparasitic strategies employed by *E. abortivum* and the defense responses by the *Armillaria* host. Finally, given that *Armillaria* species are pathogens in both natural and agronomic systems, a better understanding of this interaction may lead to the development of biocontrol methods for the control of *Armillaria* root rot.

## MATERIALS AND METHODS

**Sample collection, preparation, and sequencing.** Sporocarps of *Armillaria* sp., *E. abortivum*, and the mixed-tissue carpophoroids were observed fruiting in proximity to one another on 18 September 2015 within the Baker Woodlot and Rajendra Neotropical Migrant Bird Sanctuary, Michigan State University, East Lansing, MI (42°42′56.4″ N, 84°28′34.4″ W) (collection accession no. JRH 2446). Entire sporocarps were collected, immediately flash frozen in liquid nitrogen, and subsequently stored at −80°C. At the time of processing, three biological replicates of each of the three tissue types (*Armillaria* sp. sporocarp, *E. abortivum* sporocarp, and carpophoroid) were individually ground in liquid $N_2$. Total RNA was then extracted from the ground tissue using the Qiagen RNeasy kit (Qiagen Inc., Hilden, Germany) according to the manufacturer's protocol. RNA concentration and quality for each of the samples were assessed on a DeNovix DS-11 FX spectrophotometer (DeNovix Inc., Wilmington, DE, USA) and then shipped directly to the University of Minnesota's Genomics Center (https://genomics.umn.edu). Three technical replicates were sequenced for each biological replicate. Transcriptomic and metatranscriptomic libraries were constructed with the TruSeq standard total RNA library preparation kit with Ribo-Zero ribosomal reduction following the protocol developed by Schuierer et al. (69). Nucleotide sequencing was performed on the Illumina HiSeq 2500 system (Illumina Inc., San Diego, CA, USA), and paired-end RNA sequence reads of 51 bp were generated for further analysis.

***De novo* transcriptome assembly, transcript abundance estimation, and gene expression analysis.** The quality of the raw reads was assessed using FastQC version 0.11.9 (https://www.bioinformatics.babraham.ac.uk/projects/fastqc). The range of the number of reads for each condition is as follows: *E. abortivum* sporocarps ranged from 10,584,302 to 14,473,328, *Armillaria* sporocarps ranged from 11,712,320 to 12,431,979, and the carpophoroids (containing reads from both organisms) ranged from 9,146,682 to 12,852,086. Sequencing adaptors were trimmed, and PhiX contaminants were filtered for each sample using BBDuk (https://jgi.doe.gov/data-and-tools/bbtools/bb-tools-user-guide/bbduk-guide/). Prior to transcriptome assembly, k-mer hash sizes were estimated with *khmer* (70). *De novo* assemblies were constructed independently for both *Armillaria* sp. and *E. abortivum* with Trinity version 2.11.0 (71) using the trimmed reads generated from the respective sporocarp reads. Assembly statistics for both transcriptomes were generated with QUAST version 5 (72), and transcriptome completeness was assessed by determining the percentage of sequenced BUSCOs in each (73).

The results of the *de novo* transcriptome assemblies were used as references to perform sample-specific expression analysis. The trimmed sporocarp reads from each of the nine replicates were mapped against their respective reference transcriptomes using Bowtie 2 (74) followed by calculation of

abundance estimates using RSEM (75). The trimmed carpophoroid reads were also subsequently mapped, following the same protocol as described above, to both the *Armillaria* sp. and *E. abortivum* transcriptomes. Because of the close phylogenetic relatedness between these two species, and to filter out poorly aligned reads, we retained only mapped reads for all samples that had a MAPQ (MAPping Quality) value of 30 and above, which is equivalent to reads that have a 99.9% chance of hitting the correct match. The R package edgeR (76) and "trimmed mean of m-value" (TMM) normalization (77) were used to determine differentially upregulated transcripts between (i) *Armillaria* sporocarps and carphoroids and (ii) *E. abortivum* sporocarps and carpophoroids. Transcripts were considered differentially upregulated if they had a logFC of two or greater and a false-discovery rate (FDR)-adjusted $P$ value, or $q$ value, of <0.05. All statistical analyses for the packages listed above were conducted using R version 4.0.3 (http://www.r-project.org/).

We used SAMtools (78) to determine the number of reads from the carpophoroids that mapped to our reference transcriptomes of *E. abortivum* and our particular *Armillaria* species. To understand whether the number of reads that mapped to the carpophoroids differed significantly between each fungal species, we performed an $F$ test of equality of variances and then a two-tailed $t$ test assuming unequal variance with a $P$ value of <0.05 denoting significance.

**Sporocarp transcriptome and carpophoroid metatranscriptome annotation.** We annotated the *Armillaria* sp. and *E. abortivum* transcriptomes using Trinotate version 3.2 (79). Briefly, the transcripts were translated to coding protein sequences using TransDecoder version 5.5.0 (http://transdecoder.github.io) following identification of the longest open reading frames. To identify the most likely homologous sequence data, we used blastx on the transcripts and blastp on the predicted protein sequences (80). Using the predicted protein sequences, we also ran an HMMER (81) search against the PFAM database (82) to identify conserved domains that might be suggestive of function. We also compared these results to currently curated annotation databases such as Gene Ontology (GO) (83) and Kyoto Encyclopedia of Genes and Genomes (KEGG) (84–86). We used dbCAN2 (87) to annotate the CAZymes present in both species and compared their CAZy content to other closely related Agaricales species (22, 54–56), along with antiSMASH version 5.0 (88) to identify transcripts that belong to secondary metabolite gene clusters for both *Armillaria* and *E. abortivum*.

Finally, because this work was based off *de novo* transcriptome assemblies, there was the possibility that transcripts from contaminating organisms, including insects, yeasts, and other fungi, were also present and might be confounding the results. We approached this issue multiple ways. First, we extracted all internal transcribed spacer (ITS) regions from both transcriptomes, including the regions 18S, ITS1, 5.8S, ITS2, and 28S, using ITSx (89), and identified the origin of those sequences. To verify the number of mapped reads that represent rRNA, we also calculated the proportion of reads that mapped to the transcripts that contained the ITS sequences to the total number of mapped reads. Second, because high-quality genome assemblies of *Armillaria* exist (22, 66), we used Exonerate (90) to determine how many of the gene models in our *Armillaria* transcriptome are orthologs of *A. gallica*—the *Armillaria* species with the largest number of known gene models. Finally, all genes that we report on as important to this interaction were manually verified to be from the target organism—*Armillaria* or *E. abortivum*—using the NCBI blastx tool (80) against the nonredundant protein sequences database and confirming a hit from the Physalacriaceae or Entolomataceae/Lyophyllaceae, respectively.

**Phylogenetic analysis of *Armillaria* transcripts.** In order to identify the specific species of *Armillaria* associated in this relationship, we identified BUSCOs (73) from the transcriptome of our *Armillaria* sporocarps along with other *Armillaria* and Physalacriaceae species with previously sequenced genomes (22, 66, 91, 92). We randomly selected 100 BUSCOs to reconstruct a phylogenomic tree from the six *Armillaria* specimens (22, 66), *Guyanagaster necrorhizus* (92), and *Cylindrobasidium torrendii* (91), which served as the outgroup. Protein-coding sequences were aligned using MAFFT version 7 (93), and noninformative sites and nonaligning regions were trimmed with Gblocks (94). The 100 BUSCOs were concatenated into a supermatrix with 64,436 sites. This supermatrix was used to infer a species tree and branch support using RAxML-NG (95), using a partitioned WAG+G model, where each data partition represented an individual BUSCO.

To expand on the phylogenomic analysis above, we used the representative *Armillaria* ITS sequence obtained from ITSx, and given the close relationship of our *Armillaria* species to *A. mellea*, we pulled all *A. mellea* ITS sequences from GenBank that included associated location metadata (96–110) (see Table S1 in the supplemental material). These sequences were aligned using MAFFT version 7 (93), with refinements to the alignment performed manually. RAxML-NG (95) was used to reconstruct this phylogeny. Taxa used to root this phylogeny included *Armillaria gemina*, *Armillaria sinapina*, *Armillaria puiggarii*, and *Armillaria luteobubalina*—all members of the sister lineage to *A. mellea fide* (16).

**Data and code availability.** The raw reads and assembled transcriptomes generated during this study have been deposited in NCBI's Gene Expression Omnibus and are accessible through GEO series accession number GSE183699 or under the NCBI BioProject accession no. PRJNA761704. All other associated data, including analysis code and information on voucher collections, are available at https://github.com/HerrLab/Koch_Arma-Ento_2021.

## SUPPLEMENTAL MATERIAL

Supplemental material is available online only.

**TABLE S1**, DOCX file, 0.02 MB.

## ACKNOWLEDGMENTS

We give special thanks to Ben Lemmond and Eva Skific for providing photos of this system and to Greg Bonito at Michigan State University for generously storing sporocarp and carpophoroid tissue until we were able to transport the tissue to our laboratory. We also thank three anonymous reviewers who provided feedback and comments on the initial submission of the manuscript.

This work was completed using the Holland Computing Center of the University of Nebraska, which receives support from the Nebraska Research Initiative. This research was directly supported by startup funding from the University of Nebraska Agricultural Research Division and the University of Nebraska Office of Research and Economic Development. Additionally, J.R.H. acknowledges funding from the U.S. National Air and Space Administration (grant 80NSSC17K0737), the U.S. National Science Foundation (EPSCoR grant 1557417), and the U.S. National Institute of Justice (grant 2017-IJ-CX-0025), all of which indirectly supported this research through the support of research in his laboratory. Funding agencies had no role in study design, data collection and interpretation, or the decision to submit the work for publication.

On behalf of all authors, the corresponding author states that there is no conflict of interest.

J.R.H. initiated the work and sampled field collections; R.A.K. extracted RNA from all the tissue samples and performed the laboratory work; R.A.K. and J.R.H. processed the experimental data, analyzed the data, designed the figures, and drafted the manuscript.

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
