## [Reviewer comments · mSystems]

Transcriptomics reveals the putative mycoparasitic strategy of the mushroom *Entoloma abortivum* on species of the mushroom *Armillaria*

Rachel Koch and Joshua Herr

Corresponding Author(s): Joshua Herr, University of Nebraska - Lincoln

Review Timeline:

Submission Date:	April 30, 2021
Editorial Decision:	June 18, 2021
Revision Received:	September 20, 2021
Accepted:	September 23, 2021

Editor: Jeffrey Blanchard

Reviewer(s): Disclosure of reviewer identity is with reference to reviewer comments included in decision letter(s). The following individuals involved in review of your submission have agreed to reveal their identity: Todd Osmundson (Reviewer #2); Dan Lindner (Reviewer #3)

Transaction Report:

DOI: <https://doi.org/10.1128/mSystems.00544-21>

June 18, 2021

Dr. Joshua R Herr
University of Nebraska - Lincoln
Department of Plant Pathology
422 Plant Sciences Hall
University of Nebraska
Lincoln, NE 68503

Re: mSystems00544-21 (Transcriptomics reveals the mycoparasitic strategy of the mushroom *Entoloma abortivum* on species of the mushroom *Armillaria*)

Dear Dr. Joshua R Herr:

Thank you for submitting your manuscript to mSystems. We have completed our review and I am pleased to inform you that, in principle, we expect to accept it for publication in mSystems. However, acceptance will not be final until you have adequately addressed the reviewer comments.

All 3 reviewers suggested minor modifications to the manuscript. In sum they are substantial. Please give thought and time to addressing these comments. Once we receive the revised manuscript a decision will be made on whether further review is necessary.

Thank you for the privilege of reviewing your work. Below you will find instructions from the mSystemseitorial office and comments generated during the review.

Preparing Revision Guidelines

For complete guidelines on revision requirements, please see the Instructions to Authors at <https://msystems.asm.org/sites/default/files/additional-assets/mSys-ITA.pdf>. **Submissions of a paper that does not conform to mSystems guidelines will delay acceptance of your manuscript.**

Corresponding authors may join or renew ASM membership to obtain discounts on publication fees.

Need to upgrade your membership level? Please contact Customer Service at Service@asmusa.org.

Sincerely,

Jeffrey Blanchard

Editor, mSystems

Journals Department
Reviewer comments:

Reviewer #1 (Comments for the Author):

This research article details the transcriptional differences between fruiting bodies of two mushroom species compared to their parasitic interaction (carpophoroid). Very few studies have looked at mycoparasitic interactions, especially outside of *Trichoderma*, and this study enhances that body of work considerably. It is also the first mycoparasite transcriptome from the phylum Basidiomycota, and although the authors find many similarities to transcriptional work done on Ascomycota mycoparasites, they also find a suite of genes that together provide a new hypothesis about what mycoparasitism looks like in *Entoloma*.

Comments:

Fig 2:

-What is most striking to me is how much larger the *Armillaria* that comes from the carpophoroid is. It's has to be close to 2X as large as the other sequenced *Armillaria*. Do you think this is an error? Could it be possible that *Entoloma* or bacterial reads are being included in the *Armillaria* carpophoroid transcriptome assembly/ gene counts? The two published *A. mellea* genomes in JGI are between 14,400 - 15,700 genes. Also, there is no real discussion of this discrepancy in size in the text or what might be the cause of such a difference.

-In 2B, I think it would be helpful to change colors here to highlight 1) the data points you've generated and 2) not focus on *Armillaria* vs. all other taxa.

Fig 5: I don't believe TMM is defined anywhere in the legend.

Text:

28: You can say Ascomycota or ascomycete (not Ascomycete or Ascomycetes), ditto with Basids 134 and 152-153: "a large number" I think this is vague - How many? or What percent? Surely something that is easily put in parentheses in the main text or a small supp. table.

188: "Opposingly, two..." to "Two..."

215-217: or that the genes involved in producing the carpophoroid structure aren't that different than those involved in fruiting body formation? You don't have RNA from mycelium or even sclerotia, so we can really only say this comparison is between the *Entoloma* growing on the *Armillaria* (representing mycoparasitism and potentially some structural components?) and the *Entoloma* producing its mushroom (which is reproductive and the most complex structural composition these fungi make, but we can't say it's like actively EcM or saprobic in that tissue right?).

Side note about sclerotia: *Entoloma* is one of the many fungi that make sclerotia. I've grown this species in lab before and it makes abundant sclerotia on the plate - it's like you've got *Sclerotinia* growing or something. I wonder if the sclerotia are the main way that *Entoloma* is "hanging out" dormant until it's approximately time for *Armillaria* to start fruiting.

240-242: I'm confused here, because you say that the lectins are made by the host (other fungus) that *Trichoderma* (mycoparasite) is attacking, but then later on (253) you say maybe the lectins of *E. abortivum* (mycoparasite) are recognizing the host (*Armillaria*). I'm not saying lectins aren't involved in recognition of the host - there is definitely a wealth of literature on human C-type lectins recognizing fungi in the human body - just that the story here would be the inverse of what you say was seen in *Trichoderma*. I was confused why you made no mention of what is known about beta trefoil lectins in fungi here, but then read the section below. Now it seems much more likely that these lectins are toxins rather than involved in host recognition, but that's just my opinion.

283-296: I think this is fine to leave in the discussion, but it is really difficult to know what ABC and MFS transporters are actually transporting without doing some experimentation or looking at which specific secondary metabolite gene clusters they belong to (if at all).

326-334: My understanding is that different chitinases are involved in different parts of regular cellular processes (senescence and building of new septa, growing hyphal tip, mycoparasitism, etc.) (here's one of the *Trichoderma* references on this: Seidl, V., Huemer, B., Seiboth, B. and Kubicek, C.P., 2005. A complete survey of *Trichoderma* chitinases reveals three distinct subgroups of family 18 chitinases. *The FEBS journal*, 272(22), pp.5923-5939.). It's likely that as seen in *Trichoderma* and *Tolyocladium*, certain chitinases are only upregulated during specific conditions. It seems like the two chitinases you have identified as upregulated in the carpophore could be involved in breaking down the host tissue.

358: There is also data that mycoparasites in general are relatively reduced in CAZy content compared to other ecologies (e.g. Ahrendt et al. 2018), although there are exceptions to this when they are saprobes as well (e.g. *Trichoderma*).

402: In the future, it would be better to save at least part of a specimen from your collections to be deposited in an herbarium, so if anyone (maybe you) wants to come back to that particular collection, they can.

424: I think the numbers for the *Entoloma* carpophoroid are missing here.

494-497: I assume these accessions will be added prior to publication.

Reviewer #2 (Comments for the Author):

The manuscript by Koch and Herr examines mechanisms of mycoparasitism in the association between the gilled fungi *Entoloma abortivum* and *Armillaria* sp. using differential expression analyses of RNASeq data. Not only is this system different from those typically studied in the mycoparasitism literature (filamentous ascomycetes), but it is one that has long captivated mycologists. Although the aborted sporocarps (carpophoroids) were once thought to occur due to *Armillaria* parasitizing *Entoloma*, detailed morphological examinations from the 1970s to early 2000s

led to the conclusion that the roles were actually reversed. However, even the most recent of these studies concluded that the roles could not be assigned with absolute certainty. The study by Koch and Herr not only appears to seal the case for *Entoloma* as the pathogen and *Armillaria* as the host, but also provides a compelling explanation for how the parasitic association occurs and how it compares to other mycoparasitic associations. The greatest weakness of the study is that it fails to capture the point where the parasitic association begins (at the basidiome primordium stage), which necessarily leads to some degree of speculation in the authors' conclusions. However, this omission (seemingly due to what material was available when specimens were collected) is not a fatal flaw, as there is still much that can be learned from the comparison of mature parasitized and non-parasitized tissues. In my comments below, I note some places where clarifications could be made, but overall I find the study to be very interesting, well conceived, the methods to be appropriate to the study's objectives, and most of the conclusions to be well founded based on the results.

Signed,
Todd Osmundson

Lines 76-80: Another example worth including is *Hypomyces* and the production of "lobster mushrooms".

Lines 84-86: A perhaps better-known example that may be worth including is *Asterophora parasitica* on *Russula basidiomata*.

Lines 137-138, 154-156, and 182 (potentially elsewhere too; please double-check): the designation of "differentially expressed in ___ tissue" isn't entirely accurate, as differential expression occurs between two samples, not within a specific sample. "Differentially upregulated" (which I assume is what the authors intend here) or "differentially downregulated" would be more accurate.

Lines 142-143: State what criteria were used to determine that a gene might be important in mycoparasitic interactions. By definition, any differentially upregulated gene in the carpophoroid could be potentially important for the parasitic interaction; was this the criteria? Or was the determination based on such genes identified in other systems? Or was another criterion used?

Line 188: The meaning of the word "oppositingly" is unclear to me in this sentence. Is it meant to simply express the contrast that these are *Armillaria* vs. *Entoloma* upregulated genes, or that these genes likely play a role in *Armillaria* countering the parasitic action of *Entoloma*? Please clarify.

Line 197: Should be "North America" rather than "North American"

Lines 213-217: I think that this conclusion is very speculative and possibly incorrect, given that the initial stage of parasitism was not sampled; it also seems quite possible that expression has returned partially to a pre-initiation equilibrium. I'd recommend either removing or qualifying this statement.

Lines 222-223: "significantly upregulated" is a bit more concise than "upregulated with statistical significance".

Lines 236-237: It would be worthwhile to mention here or later in the Discussion that a follow-up study (perhaps *in vitro*) would be helpful to clarify these mechanisms.

Lines 250-253: This sentence could be clarified a bit, as it isn't entirely clear what is known and what is being suggested. Is it known that the galactose molecules are recognized by the beta-trefoil lectins? If so, then sentence could be emended to "One possible mediator of the specificity of this interaction could be the galactose sugars on the mannose protein (only known thus far from *Armillaria* species) THAT are the means by which *E. abortivum* B-trefoil-type lectins recognize and attach to its *Armillaria* host." If the recognition is what is being hypothesized, then sentence could be emended to "One possible mediator of the specificity of this interaction could be THAT the galactose sugars on the mannose protein (only known thus far from *Armillaria* species) are the means by which *E. abortivum* B-trefoil-type lectins recognize and attach to its *Armillaria* host."

Line 269: Is it correct to characterize OA secretion as a defense mechanism, or would it be more correct to characterize it as a virulence factor?

Line 271-272: Recommend clarifying. Is the meaning of this sentence that the components of the *Armillaria* "plant-parasitic arsenal" may also be used in defense against *Entoloma*?

Lines 277-279: It isn't clear to me why hyphae in the *Armillaria* mature sporocarp would be sensing nearby *Entoloma*, or how this could be considered preemptive - doesn't infection occur in the primordium?

Lines 280-282: If oxylate decarboxylase neutralizes OA after OA is produced, it isn't clear to me how it would influence gene expression of isocitrate lyase in the host. Are you suggesting a feedback mechanism? Rather than invoking a "sophisticated strategy," I think it would be better to clarify what this strategy might be, or perhaps remove this sentence since it is quite speculative (though I agree that the downregulation of isocitrate lyase in the carpophoroids does seem counterintuitive and does warrant some explanation - could it also just be that the time for defense is mostly over for *Armillaria* at this point?).

Line 285: The case for toxin extrusion by the parasite would be stronger if there were evidence for toxin production by the host; is there anything in the data that might suggest that?

Line 326: How does this number of chitinases compare to other members of Physalacreaeae?

Line 357: Conclusion of the sentence seems to be missing; recommend adding to the end of the sentence something like ", suggesting that our transcriptome is relatively complete and is therefore unlikely to underrepresent entire gene families."

Lines 377-380: I think it would be worthwhile to discuss other *Armillaria* species that have been previously implicated. From Lindner et al (reference 14):
"...from carpophoroids was also completed. Of 10 isolates from southern Wisconsin tested using the dip-hap pairing protocol of Rizzo and Harrington (1992), all were identified as *A. gallica*. Two isolates from carpophoroids collected in New Jersey (Isolate DLC99-3 and DLC99-4) were also identified as *A. gallica*, while one isolate from lower Michigan (Isolate DLC99-2) was identified as *A. ostoyae*. In addition, we have observed carpophoroid formation in clusters of *A. tabescens* basidiomes in nature, and carpophoroid formation near clusters of *A. mellea* s. s. fruiting bodies, although it is still unclear whether *A. mellea* s. s. can enter into this relationship. In work by Cha and Igarashi (1996) in Hokkaido, Japan, three isolates of *Armillaria* found associated with *E. abortivum* were identified to species. One isolate was identified as *A. gallica*, while the other two were identified as *A. jezoensis* Cha and Igarashi."

Note that this reference does not identify *A. mellea* as a host as is specified in lines 377-378 - in contrast, it specifies that it is still unclear whether that species is involved.

Line 425: Clarify parameters: which contaminants were targeted? PhiX only? This point also raises the question of how it was determined that transcripts did not originate from contaminating organisms. I assume that polyA enrichment was used, so bacteria are not likely to be a concern, but what about contaminating molds, or insects? This determination is perhaps tricky without a reference genome, and mapping to the de novo transcriptome doesn't resolve the issue, but perhaps the use of representative organisms using fastqscreen (https://www.bioinformatics.babraham.ac.uk/projects/fastq_screen/) could be used. Or, perhaps more straightforward would be to look for evidence of other ITS sequences using the method used in the manuscript to pull ITS sequences for the ITS gene tree. On another note, obtaining the ITS sequences suggests that not all ribosomal RNA was removed by the depletion step in library prep; therefore, I think it would be a good idea to specify how many of the reported reads are mRNA (or that these numbers reflect filtered data if that is the case).

Lines 494-497: Accession numbers are missing.

Line 498: URL appears to be incorrect; I believe it should be https://github.com/HerrLab/Koch_Arma-Ento_2021.

Lines 567, 571, 627: italics missing in generic name and/or specific epithet

Line 731: Capitalization missing in "zyme, e"

Figure 2: In part B, explain color of symbols; denote which are from this study.

Figure 3: Colors are a bit difficult to read in the narrower bars; consider rescaling or making lines thinner.

Figures 3-5: In Figs 3-5, the two shades of red/pink used to distinguish sporocarp and carpophoroid of *Entoloma abortivum* are difficult to distinguish.

Figure 7: Consider changing alpha value so all overlapping symbols can be seen.

Reviewer #3 (Comments for the Author):

The authors produce the first transcriptomic data from *Armillaria* and *Entoloma* fruiting bodies and carpophoroids in order to investigate whether there are signals of mycoparasitism. This work is important for understanding putative mycoparasitic interactions among Basidiomycota, where very little work has been done. Greater knowledge of mycoparasites is important ecologically, but also potentially important for understanding and controlling fungal diseases, which are increasing in importance.

I will start with some general comments, and then give specific comments.

Although the manuscript is well written and includes a good introduction, I think there are two general issues that need to be addressed in the introduction/discussion. The first is to be careful

(especially in the title) to qualify terms such as "mycoparasitic" with terms like "putatively". Although the evidence for mycoparasitism is mounting in this system, this work (and previous work) is not definitive, in my opinion. I think it would also be interesting to explore use of the term "fungicolous", as it is a term that indicates a fungus growing on another fungus, without implying the ecological relationship (parasitic, mutualistic, etc). In particular I would review the chapter "Fungicolous Fungi" by W. Gams et al in the book Biodiversity of Fungi (2004). Although a bit dated, that chapter gives a comprehensive overview of known examples of fungi growing on other fungi. I think terminology is important because there are certain things that are known (e.g. Entoloma is much more active in the carpophoroids, some genes are upregulated and others downregulated, and, based on previous work, the carpophoroids have structural similarities to Armillaria fruiting bodies, invaded by Entoloma hyphae, rather than being Entoloma fruiting bodies invaded by Armillaria). Mycoparasitism and direction of mycoparasitism are topics where evidence is mounting, but I still think it's important to be cautious with terminology. Studies showing nutrient transfer, etc would be wonderful, and that could be addressed in the discussion, although such studies are difficult to conduct, in the lab or in the field.

Another general comment is that some important groups of Basidiomycota mycoparasites are missing from the intro/discussion. The chapter by Gams et al can give a good place to start for important groups, but in particular I would draw attention to the lack of discussion of Tremella species. Tremelloid fungi are widely reported to be mycoparasites, they are grown commercially (with the host fungus), they produce structures such as haustoria, and some work has been done with transcriptomics. I think it's very important to look at this work, rather than only focusing on Ascomycota systems for comparison.

Smaller comments:

Title: As mentioned before, I would consider adding "putative" before "mycoparasitic" or modify in some way. The title is also a bit repetitive with "of the mushroom" used twice. I know the idea is to capture that both species produce mushrooms, but maybe this could be done in a less repetitive way.

Running title: I would change "of" to "and". The way it stands, it sounds like Entoloma is the host.

Line 30: I would extend the sentence to say "...mushroom-forming species and these have rarely been investigated".

Line 58: replace "kill" with "parasitize"

Line 99 to 105: Maybe modify the text a bit, as this seems quite close to the text from citation 14.

Line 188: I would strike "Opposingly", as this isn't really in opposition.

Line 204: replace "shown" with "suggested"

Line 237: just a comment to say that I agree that "sensing" probably occurs much, much earlier. It would be great to see experiments with younger carpophoroids, or to sample as the mycelia interact in culture.

Line 284: maybe excreting vs extruding?

Line 316: could these lectin be isolated? It would be very interesting to see if these really function as toxins and potential fungicides.

Line 351: in cases where you use *E. abortivum* and *E. weberi* together, I believe the genus needs to be spelled out again when switching from one genus with the same abbreviation to another genus with the same abbreviation. Journal style should be able to confirm.

Line 353: you could mention that this type of "slimmed down" genome with a reduction in CAZymes has been seen in diverse systems, even in the bat white-nose system, where the fungus *Pseudogymnoascus destructans* shows similar signs of a specialized/parasitic lifestyle, although on an animal, not a fungal host.

Around line 366: The hypothesis that *Entomolae* are parasites of "true" ectomycorrhizae is fascinating. It should also be noted that there are many species in the *Entolomatoid* clade that are mycoparasites or suspected mycoparasites (see *Claudopus/Entoloma parasiticus/parasiticum*). Some are even considered parasites of *Agaricus bisporus* growing beds. These species could be mentioned, along with the need to sequence more genomes in this clade.

Line 374. When talking about *Armillaria mellea*, it would be good to distinguish "*A. mellea* s.s." when it has really been identified as such, as opposed to previous work, such as that of Watling (1974) where that name was used, but it's hard to say which species he was working with. It might also be good to mention the species of *Armillaria* that have been confirmed to be associated with *E. abortivum*.

Line 377. The authors state that *A. mellea* has been reported before, but I believe for citation 11, this is hard to interpret, and for citation 14, it was observational (not confirmed with DNA).

Fig 2. Explain why *A. mellea* not included?

Fig. 3. For the Venn diagrams, maybe list raw numbers as well as percentages (since *Armillaria* has so many more in total, it might be good to see % as well as raw numbers).

Fig. 4. Give a better description of what volcano plots represent.

Fig. 6 is kind of small to be a stand alone figure, and I admit that although I understand ordinations, I don't really understand what is being presented in Fig. 7. With the amount of overlap, it's difficult to tell what is going on, and hard to determine what each cluster of points represents.

Re: mSystems00544-21 (Transcriptomics reveals the mycoparasitic strategy of the mushroom *Entoloma abortivum* on species of the mushroom *Armillaria*)

Dear Jeffrey and reviewers,

First of all, we want to thank you all for your time in providing feedback for our manuscript. All the comments have contributed significantly to our revision of this manuscript. Your comments were all very thoughtful and have helped us clarify our communication and add some very important details.

In the following text, we outline our responses to your comments (which are in red):

Reviewer comments:

Reviewer #1 (Comments for the Author):

This research article details the transcriptional differences between fruiting bodies of two mushroom species compared to their parasitic interaction (carpophoroid). Very few studies have looked at mycoparasitic interactions, especially outside of *Trichoderma*, and this study enhances that body of work considerably. It is also the first mycoparasite transcriptome from the phylum Basidiomycota, and although the authors find many similarities to transcriptional work done on Ascomycota mycoparasites, they also find a suite of genes that together provide a new hypothesis about what mycoparasitism looks like in *Entoloma*.

Comments:

Fig 2:

-What is most striking to me is how much larger the *Armillaria* that comes from the carpophoroid is. It's has to be close to 2X as large as the other sequenced *Armillaria*. Do you think this is an error? Could it be possible that *Entoloma* or bacterial reads are being included in the *Armillaria* carpophoroid transcriptome assembly/ gene counts? The two published *A. mellea* genomes in JGI are between 14,400 - 15,700 genes. Also, there is no real discussion of this discrepancy in size in the text or what might be the cause of such a difference.

Thank you so much for this really good comment. So we have addressed this now. First, we ran Exonerate to determine which of the gene models are *Armillaria* orthologs. Here we determined that only about 30,000 of the gene models are actually *Armillaria* in origin. The others are likely from a contaminating yeast, or represent duplicated genes with splice variants, which is a common artifact from transcriptome sequencing. We have further discussed this in the results/discussion (see L847-860).

-In 2B, I think it would be helpful to change colors here to highlight 1) the data points you've generated and 2) not focus on *Armillaria* vs. all other taxa.

Thank you! This has been done.

Fig 5: I don't believe TMM is defined anywhere in the legend.

L1883-1884. Done, thank you!

Text:

28: You can say Ascomycota or ascomycete (not Ascomycete or Ascomycetes), ditto with Basids

L28: Thanks for pointing out our capitalization error – we've fixed this in the manuscript.

134 and 152-153: "a large number" I think this is vague - How many? or What percent? Surely something that is easily put in parentheses in the main text or a small supp. table.

Thanks for catching our vague wording here. We have removed it all

188: "Opposingly, two..." to "Two..."

L399: All of the reviewers pointed this out to us and we agree and have fixed the text according.

215-217: or that the genes involved in producing the carpophoroid structure aren't that different than those involved in fruiting body formation? You don't have RNA from mycelium or even sclerotia, so we can really only say this comparison is between the *Entoloma* growing on the *Armillaria* (representing mycoparasitism and potentially some structural components?) and the *Entoloma* producing its mushroom (which is reproductive and the most complex structural composition these fungi make, but we can't say it's like actively EcM or saprobic in that tissue right?).

This is an important point and there are so many different possibilities now thinking about it more in hindsight. We ultimately decided to remove this section because we touch on evidence of mycoparasitism in *E. abortivum* later in the transcriptomics section, which is firmer than just this observation.

Side note about sclerotia: *Entoloma* is one of the many fungi that make sclerotia. I've grown this species in lab before and it makes abundant sclerotia on the plate - it's like you've got *Sclerotinia* growing or something. I wonder if the sclerotia are the main way that *Entoloma* is "hanging out" dormant until it's approximately time for *Armillaria* to start fruiting.

Thank you for your comment here. We have recorded this phenomenon in culture before and have added a short mention of it in the discussion section (L832-847).

240-242: I'm confused here, because you say that the lectins are made by the host

(other fungus) that *Trichoderma* (mycoparasite) is attacking, but then later on (253) you say maybe the lectins of *E. abortivum* (mycoparasite) are recognizing the host (*Armillaria*). I'm not saying lectins aren't involved in recognition of the host - there is definitely a wealth of literature on human C-type lectins recognizing fungi in the human body - just that the story here would be the inverse of what you say was seen in *Trichoderma*. I was confused why you made no mention of what is known about beta trefoil lectins in fungi here, but then read the section below. Now it seems much more likely that these lectins are toxins rather than involved in host recognition, but that's just my opinion.

Thank you so much for pointing out your confusion, and we think we have clarified these aspects considerably. So the first aspect we clarified was just to say lectins are known to be important for recognition in mycoparasitic interactions, and that they have been shown to be produced by both the host and the prey (so hopefully removing the confusion as to whether this is the inverse of *Trichoderma*) (L522-526). Secondly, we added a sentence about these specific lectins and their importance in recognition in mushrooms (L525-526). Lastly, in the aspect about these lectins also playing a role in toxin production, we clarify that in mushrooms, lectins function in both recognition and as a toxin, so it is not just one or the other (L686-716).

283-296: I think this is fine to leave in the discussion, but it is really difficult to know what ABC and MFS transporters are actually transporting without doing some experimentation or looking at which specific secondary metabolite gene clusters they belong to (if at all).

We left it in the manuscript but have edited the wording, but agree that it is speculative.

326-334: My understanding is that different chitinases are involved in different parts of regular cellular processes (senescence and building of new septa, growing hyphal tip, mycoparasitism, etc.) (here's one of the *Trichoderma* references on this: Seidl, V., Huemer, B., Seiboth, B. and Kubicek, C.P., 2005. A complete survey of *Trichoderma* chitinases reveals three distinct subgroups of family 18 chitinases. The FEBS journal, 272(22), pp.5923-5939.). It's likely that as seen in *Trichoderma* and *Tolyposcladium*, certain chitinases are only upregulated during specific conditions. It seems like the two chitinases you have identified as upregulated in the carpophore could be involved in breaking down the host tissue.

L731-733: This is a good point! We addressed the alternate possibility here that the low abundance of these chitinases may be due to the fact that most of the host tissue has already been broken down.

358: There is also data that mycoparasites in general are relatively reduced in CAZy content compared to other ecologies (e.g. Ahrendt et al. 2018), although there are exceptions to this when they are saprobes as well (e.g. *Trichoderma*).

This is an awesome point (and thank you for the source), and we broadened this statement out to include obligate mycoparasites and animal pathogens (L790-794).

402: In the future, it would be better to save at least part of a specimen from your collections to be deposited in an herbarium, so if anyone (maybe you) wants to come back to that particular collection, they can.

Agreed: our voucher material is currently being accessioned at the University of Michigan, so we will have an accession to include soon! Information is posted on our github repository for this study.

424: I think the numbers for the *Entoloma* carpophoroid are missing here.

These were the total reads from the carpophoroids, which is a mixed sample (*Armillaria* and *E. abortivum*), meaning the reads are from both target organisms. To clarify this, we added a parenthetical to remind readers that the carpophoroids represent both target organisms (L990).

494-497: I assume these accessions will be added prior to publication.

Yes, they are there now! See L1144-1149.

Reviewer #2 (Comments for the Author):

The manuscript by Koch and Herr examines mechanisms of mycoparasitism in the association between the gilled fungi *Entoloma abortivum* and *Armillaria* sp. using differential expression analyses of RNASeq data. Not only is this system different from those typically studied in the mycoparasitism literature (filamentous ascomycetes), but it is one that has long captivated mycologists. Although the aborted sporocarps (carpophoroids) were once thought to occur due to *Armillaria* parasitizing *Entoloma*, detailed morphological examinations from the 1970s to early 2000s led to the conclusion that the roles were actually reversed. However, even the most recent of these studies concluded that the roles could not be assigned with absolute certainty. The study by Koch and Herr not only appears to seal the case for *Entoloma* as the pathogen and *Armillaria* as the host, but also provides a compelling explanation for how the parasitic association occurs and how it compares to other mycoparasitic associations. The greatest weakness of the study is that it fails to capture the point where the parasitic association begins (at the basidiome primordium stage), which necessarily leads to some degree of speculation in the authors' conclusions. However, this omission (seemingly due to what material was available when specimens were collected) is not a fatal flaw, as there is still much that can be learned from the comparison of mature parasitized and non-parasitized tissues. In my comments below, I note some places where clarifications could be made, but overall I find the study to be very interesting, well conceived, the methods to be appropriate to the study's objectives, and most of the conclusions to be well founded based on the results.

Signed,
Todd Osmundson

Lines 76-80: Another example worth including is Hypomyces and the production of "lobster mushrooms".

L91-93: Thank you! We included this example. This is also a good segue into mushrooms!

Lines 84-86: A perhaps better-known example that may be worth including is Asterophora parasitica on Russula basidiomata.

Thank you! This has been done; see L101-102.

Lines 137-138, 154-156, and 182 (potentially elsewhere too; please double-check): the designation of "differentially expressed in ___ tissue" isn't entirely accurate, as differential expression occurs between two samples, not within a specific sample. "Differentially upregulated" (which I assume is what the authors intend here) or "differentially downregulated" would be more accurate.

This is an important point – we have updated the text in the manuscript to be more clear on this.... And we think we caught them all...

Lines 142-143: State what criteria were used to determine that a gene might be important in mycoparasitic interactions. By definition, any differentially upregulated gene in the carpophoroid could be potentially important for the parasitic interaction; was this the criteria? Or was the determination based on such genes identified in other systems? Or was another criterion used?

Thank you for pointing this out – we have updated the text to fix our omission of the text (L322-324).

Line 188: The meaning of the word "oppositingly" is unclear to me in this sentence. Is it meant to simply express the contrast that these are Armillaria vs. Entoloma upregulated genes, or that these genes likely play a role in Armillaria countering the parasitic action of Entoloma? Please clarify.

All of the other reviewers brought up the exact same point regarding this sentence, so we have reworded that sentence to be clearer and hopefully clarified the context for the reader (L399).

Line 197: Should be "North America" rather than "North American"

L439: Thank you for catching this, this has been fixed in the text.

Lines 213-217: I think that this conclusion is very speculative and possibly incorrect, given that the initial stage of parasitism was not sampled; it also seems quite possible that expression has returned partially to a pre-initiation equilibrium. I'd recommend either removing or qualifying this statement.

We ended up removing it as it is totally speculative, and the main point we were trying to make about about *E. abortivum* and mycoparasitism as part of its natural history is actually better addressed using CAZy data, so we focused on this in that section instead. We also added a few sentences in the conclusions to stress that this study, being restricted to the field collections we had, is a single time point in this interaction and not representative of the entire interaction.

Lines 222-223: "significantly upregulated" is a bit more concise than "upregulated with statistical significance".

We have updated this in the text – your suggestion improves the clarity of the sentence

Lines 236-237: It would be worthwhile to mention here or later in the Discussion that a follow-up study (perhaps in vitro) would be helpful to clarify these mechanisms.

L896-952: We have added a few sentences about future studies to more finely address this system.

Lines 250-253: This sentence could be clarified a bit, as it isn't entirely clear what is known and what is being suggested. Is it known that the galactose molecules are recognized by the beta-trefoil lectins? If so, then sentence could be emended to "One possible mediator of the specificity of this interaction could be the galactose sugars on the mannose protein (only known thus far from *Armillaria* species) THAT are the means by which *E. abortivum* B-trefoil-type lectins recognize and attach to its *Armillaria* host." If the recognition is what is being hypothesized, then sentence could be emended to "One possible mediator of the specificity of this interaction could be THAT the galactose sugars on the mannose protein (only known thus far from *Armillaria* species) are the means by which *E. abortivum* B-trefoil-type lectins recognize and attach to its *Armillaria* host."

Thank you for the suggestion! So yes, galactose molecules are recognized by beta-trefoil lectins (citation 20), so we changed it according to your suggestion; see L552-555.

Line 269: Is it correct to characterize OA secretion as a defense mechanism, or would it be more correct to characterize it as a virulence factor?

L560: Superb point and something we overlooked! In the *Coniothyrium minitans* system it is characterized as a virulence factor, so we have changed it all accordingly. Thank you!

Line 271-272: Recommend clarifying. Is the meaning of this sentence that the components of the *Armillaria* "plant-parasitic arsenal" may also be used in defense against *Entoloma*?

L560-565: We clarified this by saying: "Oxalic acid is a virulence factor employed by some plant pathogens, including species of *Armillaria*, to compromise the defense responses of the host plant by creating an acidic environment (28, 29). One differentially upregulated *Armillaria* gene in the sporocarps is isocitrate lyase (Fig. 5), which is involved in OA biosynthesis in other fungal pathogens (30, 31), and we suspect that *Armillaria* sporocarps may utilize OA to defend itself against the initial infection by *E. abortivum*."

Lines 277-279: It isn't clear to me why hyphae in the *Armillaria* mature sporocarp would be sensing nearby *Entoloma*, or how this could be considered preemptive - doesn't infection occur in the primordium?

Thank you for pointing this out. After reading it again, we decided to delete it because it's 100% speculation.

Lines 280-282: If oxalate decarboxylase neutralizes OA after OA is produced, it isn't clear to me how it would influence gene expression of isocitrate lyase in the host. Are you suggesting a feedback mechanism? Rather than invoking a "sophisticated strategy," I think it would be better to clarify what this strategy might be, or perhaps remove this sentence since it is quite speculative (though I agree that the downregulation of isocitrate lyase in the carpophoroids does seem counterintuitive and does warrant some explanation - could it also just be that the time for defense is mostly over for *Armillaria* at this point?).

This is another great point. So what we were trying to show is that *Armillaria* sporocarps produce OA likely as a way to defend against the initial *E. abortivum* infection, and that this is why we observed just dramatic upregulation of the two oxalate decarboxylases. We also hypothesize as to why we observe the downregulation of isocitrate lyase by *Armillaria* in the carpophoroids. See L562-616.

Line 285: The case for toxin extrusion by the parasite would be stronger if there were evidence for toxin production by the host; is there anything in the data that might suggest that?

This is another great point, and unfortunately, none that we observed. However, this might be something that could be disentangled if younger carpophoroids were sequenced. We've provided a full paragraph at the end explaining outstanding questions and what else we can look for in a time-series experiment.

Line 326: How does this number of chitinases compare to other members of Physalacreeae?

The line number here indicates that you likely meant the Entolomataceae and other closely related lineages. As of such, we included the number of chitinases for the species closely related to *E. abortivum* that have sequenced genomes. See L726-729.

Line 357: Conclusion of the sentence seems to be missing; recommend adding to the end of the sentence something like ", suggesting that our transcriptome is relatively complete and is therefore unlikely to underrepresent entire gene families."

L819-820: This is a great point and we have added some similar wording to clarify this sentence.

Lines 377-380: I think it would be worthwhile to discuss other *Armillaria* species that have been previously implicated. From Lindner et al (reference 14):
"...from carpophoroids was also completed. Of 10 isolates from southern Wisconsin tested using the dip-hap pairing protocol of Rizzo and Harrington (1992), all were identified as *A. gallica*. Two isolates from carpophoroids collected in New Jersey (Isolate DLC99-3 and DLC99-4) were also identified as *A. gallica*, while one isolate from lower Michigan (Isolate DLC99-2) was identified as *A. ostoyae*. In addition, we have observed carpophoroid formation in clusters of *A. tabescens* basidiomes in nature, and carpophoroid formation near clusters of *A. mellea* s. s. fruiting bodies, although it is still unclear whether *A. mellea* s. s. can enter into this relationship. In work by Cha and Igarashi (1996) in Hokkaido, Japan, three isolates of *Armillaria* found associated with *E. abortivum* were identified to species. One isolate was identified as *A. gallica*, while the other two were identified as *A. jezoensis* Cha and Igarashi."
Note that this reference does not identify *A. mellea* as a host as is specified in lines 377-378 - in contrast, it specifies that it is still unclear whether that species is involved.

This is a great point that we thank you for catching! So, we clarified what was actually written in the Lindner paper, and stated that this is the first time *A. mellea* has been confirmed as a host. We also discussed the other known species involved and that this is restricted to Eastern Asia and Eastern North America. Thank you for your thorough discussion of the issues here! See L873-886.

Line 425: Clarify parameters: which contaminants were targeted? PhiX only? This point also raises the question of how it was determined that transcripts did not originate from contaminating organisms. I assume that polyA enrichment was used, so bacteria are not likely to be a concern, but what about contaminating molds, or insects? This determination is perhaps tricky without a reference genome, and mapping to the de novo transcriptome doesn't resolve the issue, but perhaps the use of representative organisms using fastqscreen (https://www.bioinformatics.babraham.ac.uk/projects/fastq_screen/) could be used. Or, perhaps more straightforward would be to look for evidence of other ITS sequences using the method used in the manuscript to pull ITS sequences for the ITS gene tree. On another note, obtaining the ITS sequences suggests that not all ribosomal RNA was removed by the depletion step in library prep; therefore, I think it would be a good idea

to specify how many of the reported reads are mRNA (or that these numbers reflect filtered data if that is the case).

This is a fantastic point, and one that was difficult to address. We don't necessarily have a clear answer to this, but have done multiple different analyses to address this, and we are confident that the genes that we have indicated as being important to this interaction are from the two target organisms. First, to make things clearer during our methods, we have added text in those sections. The Schuierer et al study which we have cited has more information on the TruSeq Standard Total RNA Library and Ribo-Zero ribosomal reduction kit that we used during the study which removed bacterial, archaeal, and viral transcripts without poly-A tails. We next used the bbdutk script from the bbmap package in our materials and methods to remove adaptors and Phi-X contaminants bioinformatically. It uses a similar technique to the fastqscreen script you mentioned, but works in a k-mer fashion so it is much faster than fastqscreen which is based on Burrows-Wheeler mapping programs. Then, see L1055-1085 where we indicate the steps we took to quantify the potential contamination, and our reports on this in L307-309, L328-349 and L849-862. These steps include running ITSx on both transcriptomes to see which other contaminating organisms are present and quantifying their abundance. We also verified that all genes we point out as being important in this interaction are either from *Armillaria* or *Entoloma*.

Lines 494-497: Accession numbers are missing.

L1144-1149. We have added the accession numbers for the submitted raw data and assembly files.

Line 498: URL appears to be incorrect; I believe it should be https://github.com/HerrLab/Koch_Arma-Ento_2021.

Thanks for catching this, we were missing a letter in the url in the manuscript. This has been corrected.

Lines 567, 571, 627: italics missing in generic name and/or specific epithet

Thanks for finding these omitted italics – we have corrected these lines and checked all the references for generic and species names that were not italicized.

Line 731: Capitalization missing in "zyme, e"

The name of this individual is a pseudonym and is spelled in the way this individual has requested their "name" printed in the actual manuscript. A discussion of this pseudonym this author chose has been posted here: <https://f1000research.com/articles/4-900#article-comments>. We have just reflected the author names provided by the journal in this instance.

Figure 2: In part B, explain color of symbols; denote which are from this study.

Thank you, this has been done in the legend for Figure 2.

Figure 3: Colors are a bit difficult to read in the narrower bars; consider rescaling or making lines thinner.

Thank you for the suggestion, this has been done!

Figures 3-5: In Figs 3-5, the two shades of red/pink used to distinguish sporocarp and carpophoroid of *Entoloma abortivum* are difficult to distinguish.

Thank you for this suggestion, also. We adjusted the color for the *E. abortivum* to a lighter pink, which we think is a lot easier to distinguish.

Figure 7: Consider changing alpha value so all overlapping symbols can be seen.

We ended up removing this figure because it was not a crucial aspect to the story and not very intuitive.

Reviewer #3 (Comments for the Author):

The authors produce the first transcriptomic data from *Armillaria* and *Entoloma* fruiting bodies and carpophoroids in order to investigate whether there are signals of mycoparasitism. This work is important for understanding putative mycoparasitic interactions among Basidiomycota, where very little work has been done. Greater knowledge of mycoparasites is important ecologically, but also potentially important for understanding and controlling fungal diseases, which are increasing in importance.

I will start with some general comments, and then give specific comments.

Although the manuscript is well written and includes a good introduction, I think there are two general issues that need to be addressed in the introduction/discussion. The first is to be careful (especially in the title) to qualify terms such as "mycoparasitic" with terms like "putatively". Although the evidence for mycoparasitism is mounting in this system, this work (and previous work) is not definitive, in my opinion. I think it would also be interesting to explore use of the term "fungicolous", as it is a term that indicates a fungus growing on another fungus, without implying the ecological relationship (parasitic, mutualistic, etc). In particular I would review the chapter "Fungicolous Fungi" by W. Gams et al in the book *Biodiversity of Fungi* (2004). Although a bit dated, that chapter gives a comprehensive overview of known examples of fungi growing on other fungi. I think terminology is important because there are certain things that are known (e.g. *Entoloma* is much more active in the carpophoroids, some genes are upregulated and others downregulated, and, based on previous work, the carpophoroids have structural similarities to *Armillaria* fruiting bodies, invaded by *Entoloma* hyphae, rather than being *Entoloma* fruiting bodies invaded by *Armillaria*). Mycoparasitism and

direction of mycoparasitism are topics where evidence is mounting, but I still think it's important to be cautious with terminology. Studies showing nutrient transfer, etc would be wonderful, and that could be addressed in the discussion, although such studies are difficult to conduct, in the lab or in the field.

Another general comment is that some important groups of Basidiomycota mycoparasites are missing from the intro/discussion. The chapter by Gams et al can give a good place to start for important groups, but in particular I would draw attention to the lack of discussion of Tremella species. Tremelloid fungi are widely reported to be mycoparasites, they are grown commercially (with the host fungus), they produce structures such as haustoria, and some work has been done with transcriptomics. I think it's very important to look at this work, rather than only focusing on Ascomycota systems for comparison.

Thank you for this comment, and we think you are 100% correct. First, we did include a section on *Tremella* as a mycoparasite in the introduction (L93-95). However, we looked extensively throughout the literature for gene expression and/or transcriptomics studies of basidiomycete mycoparasites to little avail. There is a recent study out which used metatranscriptomics to understand the role Tremella plays in the wolf lichen (Two Basidiomycete Fungi in the Cortex of Wolf Lichens, Tuovinen et al. 2019, Current Biology), however, all of the upregulated genes in that system had no close hits in available databases, making it difficult to do some comparisons. We are not opposed to adding in more info, we just might not be looking in the right place. Therefore, if this reviewer has any citations for me to follow up on, we are more than happy. Moreover though, we do not think *not* having this information takes away from the manuscript, because we do show that even distantly related organisms utilize similar mechanisms to be a successful mycoparasite.

Smaller comments:

Title: As mentioned before, I would consider adding "putative" before "mycoparasitic" or modify in some way. The title is also a bit repetitive with "of the mushroom" used twice. I know the idea is to capture that both species produce mushrooms, but maybe this could be done in a less repetitive way.

This has been done.

Running title: I would change "of" to "and". The way it stands, it sounds like Entoloma is the host.

Thanks for the input here – we have updated the running title to clarify the interaction.

Line 30: I would extend the sentence to say "...mushroom-forming species and these have rarely been investigated".

L30-31: We have updated the sentence to add this clarification.

Line 58: replace "kill" with "parasitize"

L66: We have changed this.

Line 99 to 105: Maybe modify the text a bit, as this seems quite close to the text from citation 14.

L137-146: We have re-worded these sentences accordingly.

Line 188: I would strike "Opposingly", as this isn't really in opposition.

We have eliminated this word in the manuscript.

Line 204: replace "shown" with "suggested"

L441: We have updated the wording of this sentence.

Line 237: just a comment to say that I agree that "sensing" probably occurs much, much earlier. It would be great to see experiments with younger carpophoroids, or to sample as the mycelia interact in culture.

We initially brushed over the limitations of sampling and timing of finding the interactions in the field, but we have added a section to the paper to discuss this. All the reviewers commented on this, so we did not adequately cover this topic in the discussion. Thanks for bringing it to our attention. Check out L891-947 where we discuss this.

Line 284: maybe excreting vs extruding?

We agree that "excreting" is a better word here and have updated the text in the manuscript (L613).

Line 316: could these lectin be isolated? It would be very interesting to see if these really function as toxins and potential fungicides.

This is an awesome point, and beyond my own skills in molecular biology currently 😊 However, we did add a statement in the conclusions about the potential for biocontrol of this system for potentially controlling *Armillaria* infections (L947-950).

Line 351: in cases where you use *E. abortivum* and *E. weberi* together, I believe the genus needs to be spelled out again when switching from one genus with the same abbreviation to another genus with the same abbreviation. Journal style should be able to confirm.

Yes, this is a great point that adds clarity to the species names. We have added full genus names to both *Tolypocladium ophioglossoides* and *Escovopsis weberi* in the text of the manuscript.

Line 353: you could mention that this type of "slimmed down" genome with a reduction in CAZymes has been seen in diverse systems, even in the bat white-nose system, where the fungus *Pseudogymnoascus destructans* shows similar signs of a specialized/parasitic lifestyle, although on an animal, not a fungal host.

This is a great point, and we broadened this statement out to include other obligate mycoparasites and animal pathogens (L790-794).

Around line 366: The hypothesis that *Entomolae* are parasites of "true" ectomycorrhizae is fascinating. It should also be noted that there are many species in the Entolomatoid clade that are mycoparasites or suspected mycoparasites (see *Claudopus/Entoloma parasiticus/parasiticum*). Some are even considered parasites of *Agaricus bisporus* growing beds. These species could be mentioned, along with the need to sequence more genomes in this clade.

Thank you for this comment; we added that important information as well as a call for more genome sequencing of species in this clade.

Line 374. When talking about *Armillaria mellea*, it would be good to distinguish "A. mellea s.s." when it has really been identified as such, as opposed to previous work, such as that of Watling (1974) where that name was used, but it's hard to say which species he was working with. It might also be good to mention the species of *Armillaria* that have been confirmed to be associated with *E. abortivum*.

This is an important point you bring up, and, along your next comment, we have updated the text to add clarity here; see L868-878.

Line 377. The authors state that *A. mellea* has been reported before, but I believe for citation 11, this is hard to interpret, and for citation 14, it was observational (not confirmed with DNA).

Yes, this is correct. We have updated the language in these sentences to better reflect the previous literature to address the *Armillaria* taxa in this association.

Fig 2. Explain why *A. mellea* not included?

We just included it! Apologies for that oversight.

Fig. 3. For the Venn diagrams, maybe list raw numbers as well as percentages (since *Armillaria* has so many more in total, it might be good to see % as well as raw numbers).

Great suggestion, and these are now included on the venn diagram!

Fig. 4. Give a better description of what volcano plots represent.

Thank you! In the figure legend, we included the following statement: Genes with the highest statistical significance and the largest fold change will be represented by dots towards the top of the plot that are far to either the left- (carpophoroid) or right-hand (sporocarp) sides.

Fig. 6 is kind of small to be a stand alone figure, and I admit that although I understand ordinations, I don't really understand what is being presented in Fig. 7. With the amount of overlap, it's difficult to tell what is going on, and hard to determine what each cluster of points represents.

For Fig. 7, after consideration of your comment, along with those of other reviewers, we decided to remove it because you are right... it is not 100% intuitive and is not crucial to our story.

For Fig. 6, while we do agree that it is small for a stand alone figure, it is a crucial (and intuitive) part of this story, so we have opted to keep it as such for now. Unfortunately, it does not lend itself easily to combination with another figure.

September 23, 2021

Dr. Joshua R Herr
University of Nebraska - Lincoln
Department of Plant Pathology
422 Plant Sciences Hall
University of Nebraska
Lincoln, NE 68503

Re: mSystems00544-21R1 (Transcriptomics reveals the putative mycoparasitic strategy of the mushroom *Entoloma abortivum* on species of the mushroom *Armillaria*)

Dear Dr. Joshua R Herr:

Thank you for your detailed responses to the comments and the many changes you have made. Your manuscript has been accepted, and I am forwarding it to the ASM Journals Department for publication. For your reference, ASM Journals' address is given below. Before it can be scheduled for publication, your manuscript will be checked by the mSystems senior production editor, Ellie Ghatineh, to make sure that all elements meet the technical requirements for publication. She will contact you if anything needs to be revised before copyediting and production can begin. Otherwise, you will be notified when your proofs are ready to be viewed.

As an open-access publication, mSystems receives no financial support from paid subscriptions and depends on authors' prompt payment of publication fees as soon as their articles are accepted. =

Publication Fees:

We recognize that the video files can become quite large, and so to avoid quality loss ASM suggests sending the video file via <https://www.wetransfer.com/>. When you have a final version of the video and the still ready to share, please send it to Ellie Ghatineh at eghatineh@asmusa.org.

Sincerely,

Jeffrey Blanchard
Editor, mSystems

Journals Department
Table S1: Accept